# History-dependent physiological adaptation to lethal genetic modification under antibiotic exposure

**Yuta Koganezawa[1], Miki Umetani[1,2], Moritoshi Sato[2,3,4], Yuichi Wakamoto[1,2,4]***

[1]Department of Basic Science, Graduate School of Arts and Sciences, The University of Tokyo, Meguro-ku, Japan; [2]Research Center for Complex Systems Biology, The University of Tokyo, Tokyo, Japan; [3]Department of Life Sciences, Graduate School of Arts and Sciences, The University of Tokyo, Tokyo, Japan; [4]Universal Biology Institute, The University of Tokyo, Tokyo, Japan

**Abstract** Genetic modifications, such as gene deletion and mutations, could lead to significant changes in physiological states or even cell death. Bacterial cells can adapt to diverse external stresses, such as antibiotic exposure, but can they also adapt to detrimental genetic modification? To address this issue, we visualized the response of individual *Escherichia coli* cells to deletion of the antibiotic resistance gene under chloramphenicol (Cp) exposure, combining the light-inducible genetic recombination and microfluidic long-term single-cell tracking. We found that a significant fraction (~40%) of resistance-gene-deleted cells demonstrated a gradual restoration of growth and stably proliferated under continuous Cp exposure without the resistance gene. Such physiological adaptation to genetic modification was not observed when the deletion was introduced in 10 hr or more advance before Cp exposure. Resistance gene deletion under Cp exposure disrupted the stoichiometric balance of ribosomal large and small subunit proteins (RplS and RpsB). However, the balance was gradually recovered in the cell lineages with restored growth. These results demonstrate that bacterial cells can adapt even to lethal genetic modifications by plastically gaining physiological resistance. However, the access to the resistance states is limited by the environmental histories and the timings of genetic modification.

**\*For correspondence:**
cwaka@mail.ecc.u-tokyo.ac.jp

**Competing interest:** The authors declare that no competing interests exist.

## Editor's evaluation

This paper presents the temporal relationships between deletion of a resistance gene, introduction of antibiotic, and cell growth that are intriguing and novel. It will be of interest to researchers studying heterogeneity in antibiotic tolerance and the origins of drug resistance.

## Introduction

Bacteria in nature are constantly challenged by environmental and genetic perturbations and must be robust against them for survival. Bacterial cells possess programs to adapt or resist such perturbations. For example, under the conditions of nutrient deprivation, *E. coli* and related bacteria provoke RpoS-mediated general stress response and globally change the metabolic and gene expression profiles to protect themselves from the stress (*Battesti et al., 2011*). DNA damage also induces SOS response, which promotes both DNA repair and mutagenesis (*Walker, 1996*; *Taddei et al., 1995*). If the mutations in essential genetic elements remain unrepaired, their influence will be propagated to cellular phenotypes or even cause cell death. How rapidly new genetic changes alter cellular phenotypes and whether they always give rise to the same phenotypes in given environments are fundamental

questions in genetics (*Ryan, 1955*; *Sun et al., 2018*). Nevertheless, such time-dependent and redundant genotype-phenotype correspondences are usually deemed negligible or insignificant in most genetics analyses.

However, the evidence suggests that biological systems can buffer or compensate for the impact of genetic changes at the cytoplasmic and physiological levels (*Waddington, 1942*). For example, specific molecular chaperons could hinder genetic variations from manifesting as morphological and growth phenotypes in fruit flies (*Rutherford and Lindquist, 1998*), plants (*Queitsch et al., 2002*), and bacteria (*Fares et al., 2002*). Furthermore, the loss of biological functions caused by genetic changes can be compensated for by modulating the RNA and protein levels of the mutated or other related genes in many organisms (*El-Brolosy and Stainier, 2017*). In *Bacillus subtilis*, the expression noise of mutant sporulation regulator results in the partial penetrance of its influence to spore-forming phenotypes (*Eldar et al., 2009*). These observations across diverse organisms suggest that phenotypic consequences of genetic modifications can be modulated based on environmental and physiological contexts, which may promote the survival and evolution of the organisms.

Despite these experimental implications, it remains elusive whether bacterial cells can circumvent even lethal genetic modifications such as antibiotic resistance gene deletion under antibiotic exposure. Furthermore, how cells initially respond to the genetic changes and how their physiological and phenotypic states are modulated in longer timescales are poorly characterized. However, addressing these issues requires precisely defining the timings of genetic changes and tracking individual cells for a long period to unravel their phenotypic transitions and consequences (*Sun et al., 2018*).

In this study, we resolve these technical issues by combining the photoactivatable Cre (PA-Cre) light-inducible genetic recombination technique (*Kawano et al., 2016*) and microfluidic long-term single-cell tracking (*Wang et al., 2010*). We induced a pre-designed deletion of chromosomally encoded and fluorescently tagged drug resistance gene in *E. coli* directly in the microfluidic device. The results show that all of the resistance-gene-deleted cells under continuous drug exposure showed a decline in growth in 5–7 generations, but a fraction of the resistance-gene-deleted cells gradually restored their growth without additional mutations. In contrast, no cells restored growth when the same deletion was introduced 10 hr or more in advance before drug exposure. Therefore, bacterial cells can physiologically adapt to lethal genetic modifications. However, its feasibility depends on environmental histories, the timings of genetic modifications, and the severity of the antibiotic stress.

## Results

### Resistance gene deletion in *E. coli* by blue-light illumination

To investigate the response of individual cells to antibiotic resistance gene deletion, we constructed an *E. coli* strain expressing chloramphenicol acetyltransferase (CAT) tagged with mCherry red fluorescent protein (*Figure 1A*). CAT confers resistance to chloramphenicol (Cp) by acetylating Cp (*Shaw, 1967*). The *mcherry-cat* resistance gene was integrated on the chromosome along with the upstream and downstream *loxP* sequences so that the resistance gene could be excised by Cre recombinase (*Figure 1A*). We also introduced the PA-Cre recombination system, which consists of split Cre-recombinase fragments, called CreC and CreN, attached to p-Magnet (p-Mag) and n-Magnet (n-Mag) monomers, respectively (*Figure 1B*; *Kawano et al., 2015*). p-Mag and n-Mag are heterodimers derived from the fungal photo-receptor, Vivid (*Kawano et al., 2015*), which heterodimerize upon blue-light illumination. In the PA-Cre system, blue-light illumination leads to heterodimerizations of p-Mag-CreC and n-Mag-CreN fragments and recovers Cre recombination activity (*Figure 1B*; *Kawano et al., 2016*). Therefore, this system allows us to induce the resistance gene deletion at arbitrary timings by blue-light illumination. The original PA-Cre system was designed for use in mammalian cells (*Kawano et al., 2016*); we thus replaced the plasmid backbone and the promoter for use in *E. coli*.

We first analyzed the response of the constructed strain YK0083 to blue-light illumination under drug-free conditions. Single-cell observations with the mother machine microfluidic device (*Wang et al., 2010*) and custom stage-top LED illuminator (*Figure 1—figure supplement 1*) revealed that 30-min blue-light exposure ($\lambda = 464{\sim}474$ nm, 6.8 mW at the specimen position) led to the loss of mCherry-CAT fluorescence in 25% (50/200) cells (*Figure 1C and D*, and *Video 1*), suggesting the deletion of the *mcherry-cat* gene in these cell lineages. The fluorescence signals decayed to the background level in 4–5 generations, and the decay kinetics was consistent with the dilution by growth

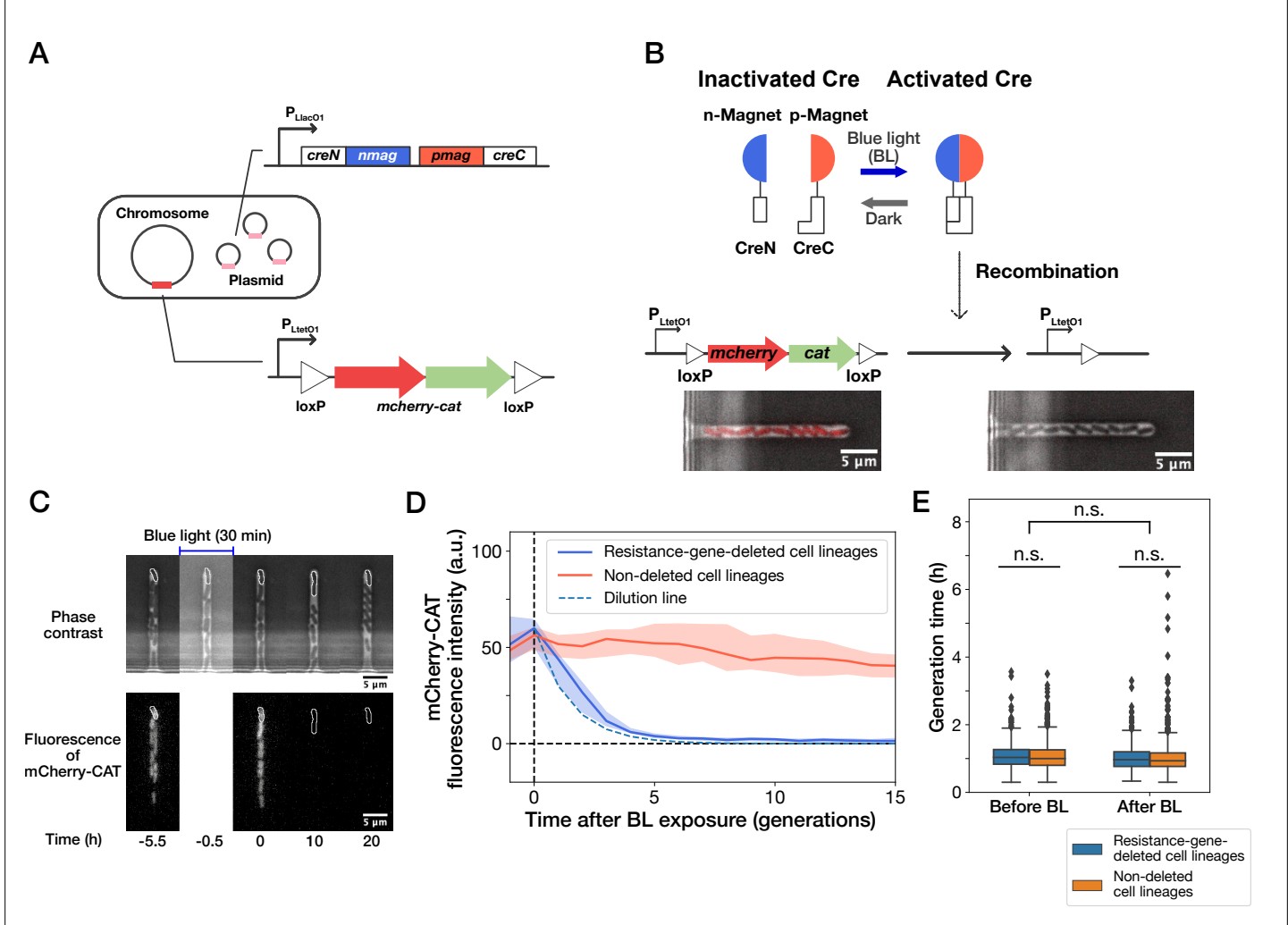

**Figure 1.** Live-cell monitoring of phenotypic transitions in response to gene deletion. (**A**) Schematic drawing of an *E. coli* strain, YK0083. This strain harbors the photo-removable *mcherry-cat* gene on the chromosome and a low-copy plasmid carrying the *pa-cre* genes (*creN-nmag* and *pmag-creC*). (**B**) The PA-Cre system. Blue-light illumination provokes the dimerization of two PA-Cre fragments and induces the deletion of the *mcherry-cat* gene. The micrographs represent the combined images of phase contrast and mCherry-CAT fluorescence channels. Left and right images show the cells before and after blue-light illumination, respectively. (**C**) Representative time-lapse images of gene-deletion experiments. The upper and lower images show the cells in phase contrast and mCherry-CAT fluorescence channels, respectively. The cell lineages at the closed end of the growth channel (outlined in white) were monitored. A 30-min blue-light illumination starting at $t = -0.5$ hr led to the loss of mCherry-CAT fluorescence signals in this cell lineage. (**D**) The transitions of mCherry-CAT fluorescence intensities in resistance-gene-deleted (blue) and non-deleted (red) cell lineages. Lines and shaded areas represent the medians and the 25–75% ranges, respectively. The cyan broken line represents the expected fluorescence decay curve when the fluorescence intensity decreases to half in each generation. (**E**) Generation time of resistance-gene-deleted and non-deleted cells 10 generations before and after blue-light illumination. The middle line and both edges of the boxes represent the medians and the 25–75% ranges of generation time. Whiskers indicate the minimum and maximum of the data except for the outliers. The points represent the outliers. No significant differences in generation time were detected between the groups at the significance level of 0.01 (p = 0.47 for before BL, p = 0.027 for after BL, p = 0.055 for before BL vs after BL, Mann-Whitney U test).

The online version of this article includes the following figure supplement(s) for figure 1:

**Figure supplement 1.** On-stage blue-light LED illuminator.

**Figure supplement 2.** Resistance gene deletion by blue-light illumination in batch cultures.

(*Figure 1D*). Furthermore, the 30 min blue-light illumination and genetic recombination did not affect the growth of individual cells (*Figure 1E*).

Extending illumination duration increased the frequency of cells showing loss of fluorescence, reaching 100% ($n = 296$) in 4 hr (*Figure 1—figure supplement 2A* and *Figure 1—figure supplement*

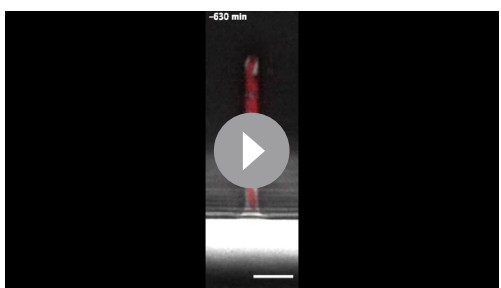

**Video 1.** Deletion of *mcherry-cat* gene by blue-light illumination under drug-free conditions. YK0083 cells were cultured in the mother machine flowing the M9 minimal medium and exposed to blue light from *t* = −30 min to 0 min (marked with ' + BL'). The merged images of phase contrast (grayscale) and mCherry-CAT fluorescence (red) channels are shown. The fluorescence of mCherry-CAT was gradually lost after blue-light illumination in this cell lineage. Scale bar, 5 μm.

https://elifesciences.org/articles/74486/figures#video1

2B). We confirmed the correspondence between the loss of mCherry fluorescence and *cat* gene deletion by colony PCR (*Figure 1—figure supplement 2C*).

## Fractional continuation of growth against resistance gene deletion under drug exposure

We next induced the deletion of the resistance gene in YK0083 cells in the mother machine, continuously flowing a medium containing 15 μg/mL of Cp. This drug concentration was 1.5-fold higher than the minimum inhibitory concentration (MIC) of the non-resistant strain YK0085 (10 μg/mL), which was constructed by illuminating the YK0083 cells by blue light in batch culture (*Figure 2*). Therefore, it was expected that this drug concentration would inhibit the growth of resistance-gene-deleted cells. This Cp concentration was significantly lower than the MIC of resistant YK0083 cells (100 μg/mL, *Figure 2*) and did not influence their elongation rates (*Figure 2—figure supplement 1*).

A 30-min blue-light illumination induced the *mcherry-cat* gene deletion in 24.5% (343/1399) in YK0083 cells (*Figure 3A* and *Video 2*). While non-deleted cells continued to grow with their generation time (interdivision time) and mCherry-CAT fluorescence intensity nearly unaffected (*Figure 3B and C*), resistance-gene-deleted cells gradually showed a decline in growth (*Figure 3D–G*). The fluorescence intensity of the resistance-gene-deleted cells decayed to the background levels in 4–5 generations (*Figure 3D, F and H*), and their generation time also increased correspondingly (*Figure 3E, G and I*). However, while the growth of 62.7% (163/260) of resistance-gene-deleted cells was eventually stopped, the other 37.3% cells restored and continued their growth over 30 generations without the *cat* resistance gene (*Figure 3H and I*). The generation time of these cell lineages recovered from 6.3 hr (the median of 6th-9th generations) to 3.0 hr (the median of 21st-30th generations) under the

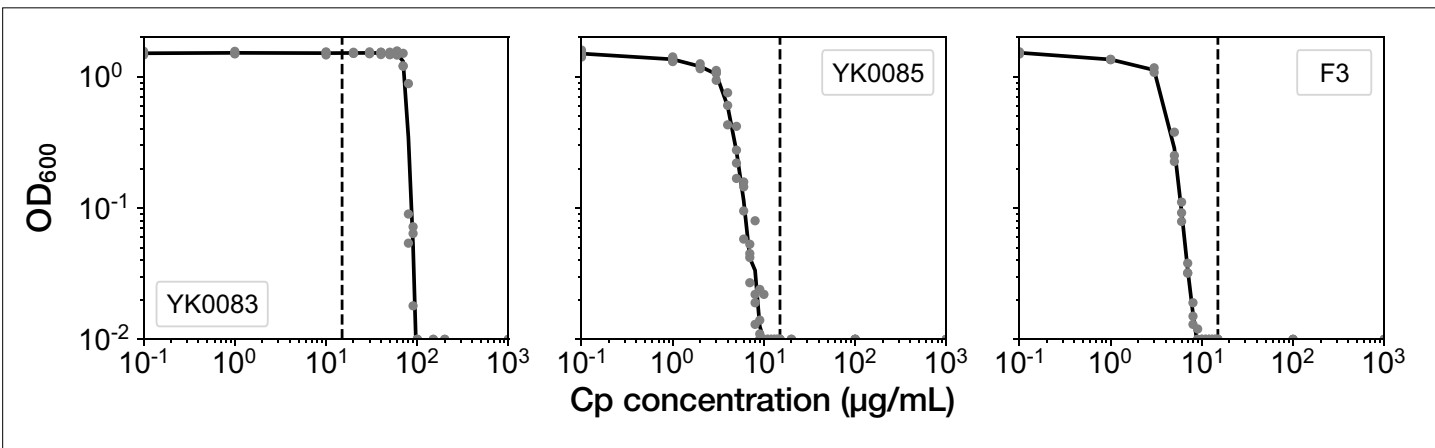

**Figure 2.** MIC tests. Gray points represent the $OD_{600}$ of the indicated strains after a 23 hr incubation period in the M9 media containing the corresponding concentrations of Cp. The minimum concentration where $OD_{600}$ became lower than 0.01 was adopted as the MIC for each strain. A total of 15 μg/mL of Cp was used in the time-lapse experiments (dashed line). The measurements were repeated at least thrice.

The online version of this article includes the following figure supplement(s) for figure 2:

**Figure supplement 1.** The Cp concentration used in the time-lapse measurements caused no significant effect on the growth of non-deleted YK0083 cells.

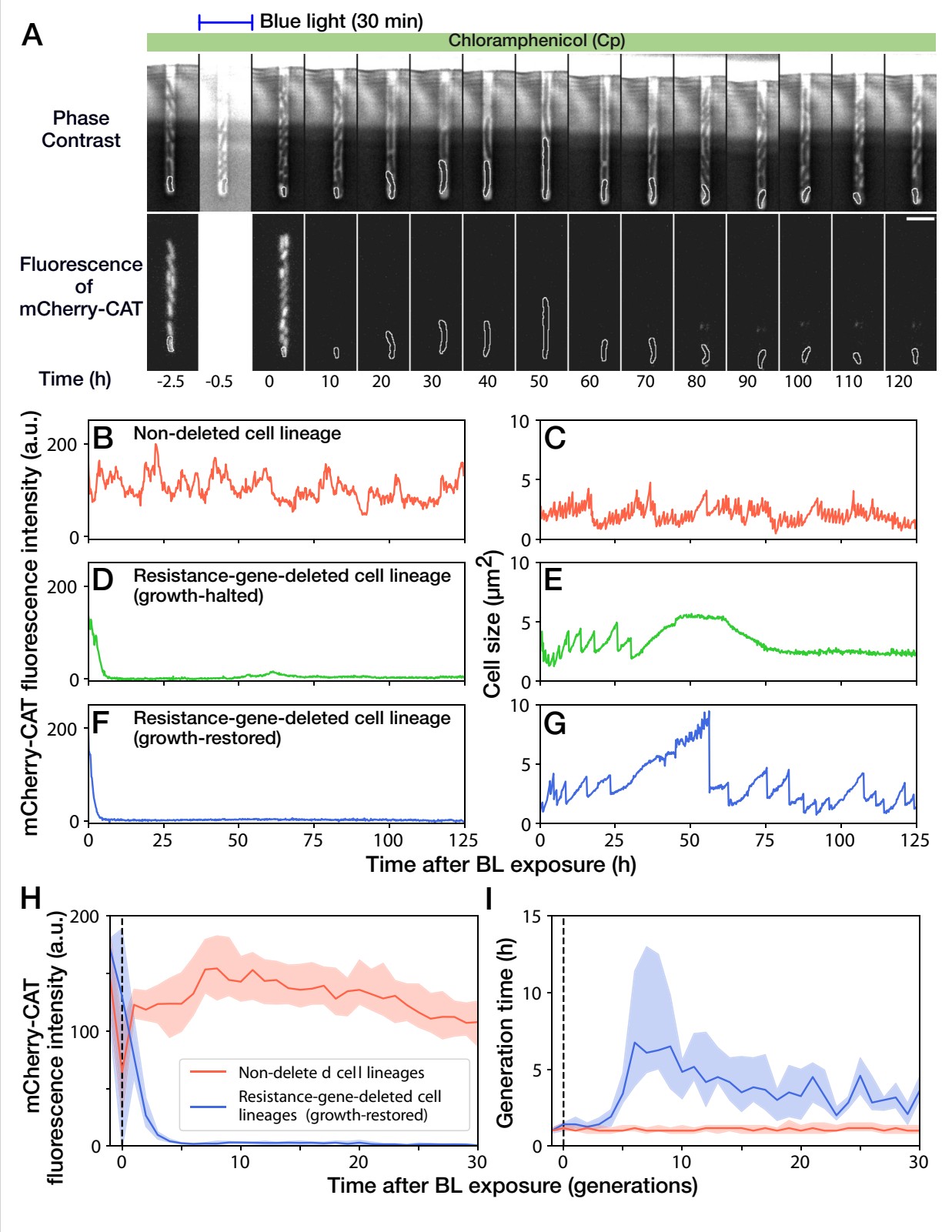

**Figure 3.** Growth continuation under Cp exposure against *cat* gene deletion. (**A**) Time-lapse images of a cell lineage that continued growth and division against *mcherry-cat* gene deletion. The upper and lower sequences show phase contrast and mCherry fluorescence images, respectively. Blue light was illuminated from *t* = -0.5 h to 0 hr. The cells at the closed end of the growth channel were monitored during the experiment, which are outlined in white on the images. Scale bar, 5 μm. (**B–G**) The transitions of mCherry-CAT fluorescence intensities and cell size in single-cell lineages. (**B, C**) Non-deleted

*Figure 3 continued on next page*

*Figure 3 continued*

cell lineage. (**D, E**) Growth-halted resistance-gene-deleted cell lineage. The decrease in cell size after 60 hr was due to the shrinkage of the cell body. (**F, G**) Growth-restored resistance-gene-deleted cell lineage. (**H, I**) The transitions of mCherry-CAT fluorescence intensities (**H**) and generation time (**I**). The lines and shaded areas represent the medians and the 25–75% ranges, respectively. Red represents non-deleted cell lineages. Blue represents growth-restored resistance-gene-deleted cell lineages. The transitions are shown in generations.

The online version of this article includes the following figure supplement(s) for figure 3:

**Figure supplement 1.** Transitions of elongation rates after blue-light illumination.

**Figure supplement 2.** Resistance gene deletion in batch cultures under Cp exposure.

**Figure supplement 3.** Correspondence between fluorescence loss and *mcherry-cat* gene deletion in the cells sampled from the mother machine.

**Figure supplement 4.** Growth recovery of resistance-gene-deleted cell lineages after Cp removal.

**Figure supplement 5.** Cellular phenotypes before blue-light illumination do not correlate with the cellular fates.

continuous drug exposure (*Figure 3I*). The elongation rate transitions across multiple generations also manifested the growth recovery (*Figure 3—figure supplement 1*).

We first suspected that the growth recovery under Cp exposure was attributed to the presence of the *cat* gene untagged from the *mcherry* gene or by additional unintended resistance-enhancing mutations. To test this hypothesis, we illuminated batch cultures of YK0083 cells by blue light for 30 min under exposure to 15 µg/mL of Cp and plated them on agar plates. Colony PCR confirmed that all the cells that lost mCherry fluorescence were also *cat*-negative even when the deletion was introduced under Cp exposure (*Figure 3—figure supplement 2A*). This result strongly suggests that the growth-restored cells do not retain the *cat*-resistance gene as designed. The fraction of resistance-gene-deleted cells after 30-min blue-light illumination monotonically decreased in batch culture containing 15 µg/mL of Cp (*Figure 3—figure supplement 2B*), which is consistent with the observation that growth-restored resistance-gene-deleted cells grew more slowly than non-deleted cells.

In addition, we sampled the culture media flowing out from the microfluidic device and obtained cell populations derived from a single or few ancestral cells by limiting dilution (*Figure 3—figure supplement 3A*). PCR analysis confirmed the lack of *cat* gene in the non-fluorescent cell populations (*Figure 3—figure supplement 3B*). Furthermore, whole-genome sequencing of the cell populations obtained by limiting dilution showed no additional mutations in four out of five non-fluorescent cell populations tested (*Table 1*). One point mutation was present in one cellular population, but this mutation did not affect the MIC (*Figure 3—figure supplement 3C*). To calculate the probability that all of these five non-fluorescent cell populations had originated from resistance-gene-deleted growth-halted cells, we analyzed the regrowing dynamics of both growth-restored and growth-halted cell lineages after removing Cp in the microfluidic device (*Figure 3—figure supplement 4A-C* and *Video 3*). We found that 91.3% (73/80) of growth-restored cell lineages recovered fast growth after the exposure to 15 µg/mL of Cp for 72 hr. On the other hand, the proportion of cells that recovered growth was lower for the growth-halted cell lineages; only 69.3% (140/202) of growth-halted cell lineages could resume growth. The growth after first cell divisions were as fast as that for the non-deleted cells and was indistinguishable between the growth-restored and growth-halted cell lineages (*Figure 3—figure supplement 4D*). Taking the fraction of growth-restored cells among the resistance-gene-deleted cells and the proportions of growth-recovered cells after Cp removal into account, we found that the probability that all the five cell populations were derived from growth-halted cell lineages was 5.5% (see

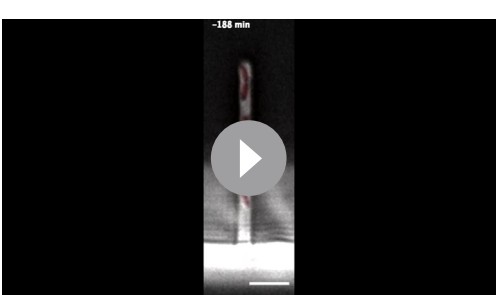

**Video 2.** Growth restoration under Cp exposure after resistance gene deletion. YK0083 cells were cultured in the mother machine flowing the M9 minimal medium containing 15 µg/mL of Cp. Blue light was illuminated from *t* = −30 min to 0 min (marked with ' + BL'). The merged images of phase contrast (grayscale) and mCherry-CAT fluorescence (red) channels are shown. The illumination caused the loss of fluorescence signals, that is, the deletion of the *mcherry-cat* gene, in this cell lineage. Nevertheless, the cell gradually restored growth and division under continuous Cp exposure. Scale bar, 5 µm.

https://elifesciences.org/articles/74486/figures#video2

**Table 1.** Mutations detected by whole-genome sequencing.
We obtained isolated cellular populations derived from single or few cells by limiting dilution of the culture media flowing out from the mother machine. We detected only one point mutation in Sample 3.

| Sample no. | Mutation | Position | Base | Annotation |
|---|---|---|---|---|
| Sample 1 | No mutation | | | |
| Sample 2 | No mutation | | | |
| Sample 3 | Mutation in leuC site | 80,471 | G -> T | A132S |
| Sample 4 | No mutation | | | |
| Sample 5 | No mutation | | | |

Materials and methods). Therefore, the probability that unintended genetic changes were responsible for the growth restoration is small.

It is also unlikely that the growth restoration was caused by the residual mCherry-CAT proteins because 30 consecutive cell divisions dilute cytoplasmic proteins $2^{30} \approx 10^9$ folds if additional production is prevented; note that even the total number of protein molecules in a bacterial cell are in the order of $10^6$-$10^7$ (**Milo, 2013**) We also detected no significant differences in the mCherry-CAT fluorescence intensity before blue-light illumination between the growth-halted and growth-restored cell lineages (**Figure 3—figure supplement 5A**). Furthermore, no significant differences were observed even between the resistance-gene-deleted and non-deleted cell lineages (**Figure 3—figure supplement 5A**). We also examined the influence of elongation rate before blue-light illumination on the gene deletion and the fates after gene deletion, finding no correlations (**Figure 3—figure supplement 5B**). Therefore, neither the amount of mCherry-CAT proteins at the time of gene deletion nor pre-deletion elongation rate affected the likelihood of gene deletion and the determination of growth-halt and growth-restoration fates under these experimental conditions.

We also found that deleting the *mcherry-cat* gene flowing a medium containing a twofold concentration of Cp (i.e., 30 μg/mL) eliminated growth-restored cell lineages: 33.1% (361/1092) of the cells illuminated by blue light lost the resistance gene, and none of them restored growth. This result suggests that deleting the resistance gene at the concentrations of Cp sufficiently higher than the MIC can prevent physiological adaptation.

## Resistance gene deletion long before Cp exposure prevents growth restoration

The high frequency of growth restoration observed against the resistance gene deletion was unexpected since the MIC of the susceptible strain was below 15 μg/mL (**Figure 2** and **Figure 3—figure**

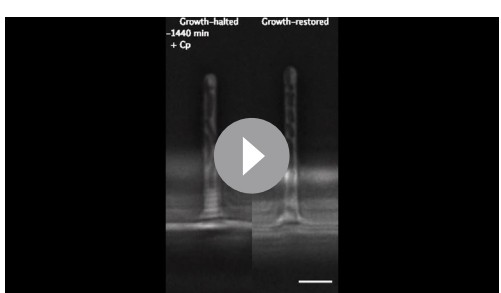

**Video 3.** Growth recovery of resistance-gene-deleted cells after Cp removal. Left, growth-halted resistance-gene-deleted cell; Right, Growth-restored resistance-gene-deleted cells. Cp was removed at *t* = 0 min (72 hr after blue light illumination). The merged images of phase contrast (grayscale) and mCherry-CAT fluorescence (red) channels are shown. Scale bar, 5 μm.
https://elifesciences.org/articles/74486/figures#video3

**supplement 3C**). To understand this discrepancy, we cultured the susceptible YK0085 cells in the mother machine first without Cp and then exposed them to 15 μg/mL of Cp directly in the device. In contrast to the previous observation, all cells stopped growth and division entirely, with no cells restoring growth ("Pre-deleted" in **Figure 4A–C** and **Video 4**). This result excludes the hypothesis that the unique cultivation environments in the microfluidics device are the cause of growth restoration. Instead, this observation implies that the timing of gene deletion is crucial for the cells to withstand the Cp exposure without the resistance gene and restore growth.

To further investigate the importance of the timing of gene deletion, we performed microfluidic single-cell measurements with the resistant YK0083 strain, varying the duration from blue-light

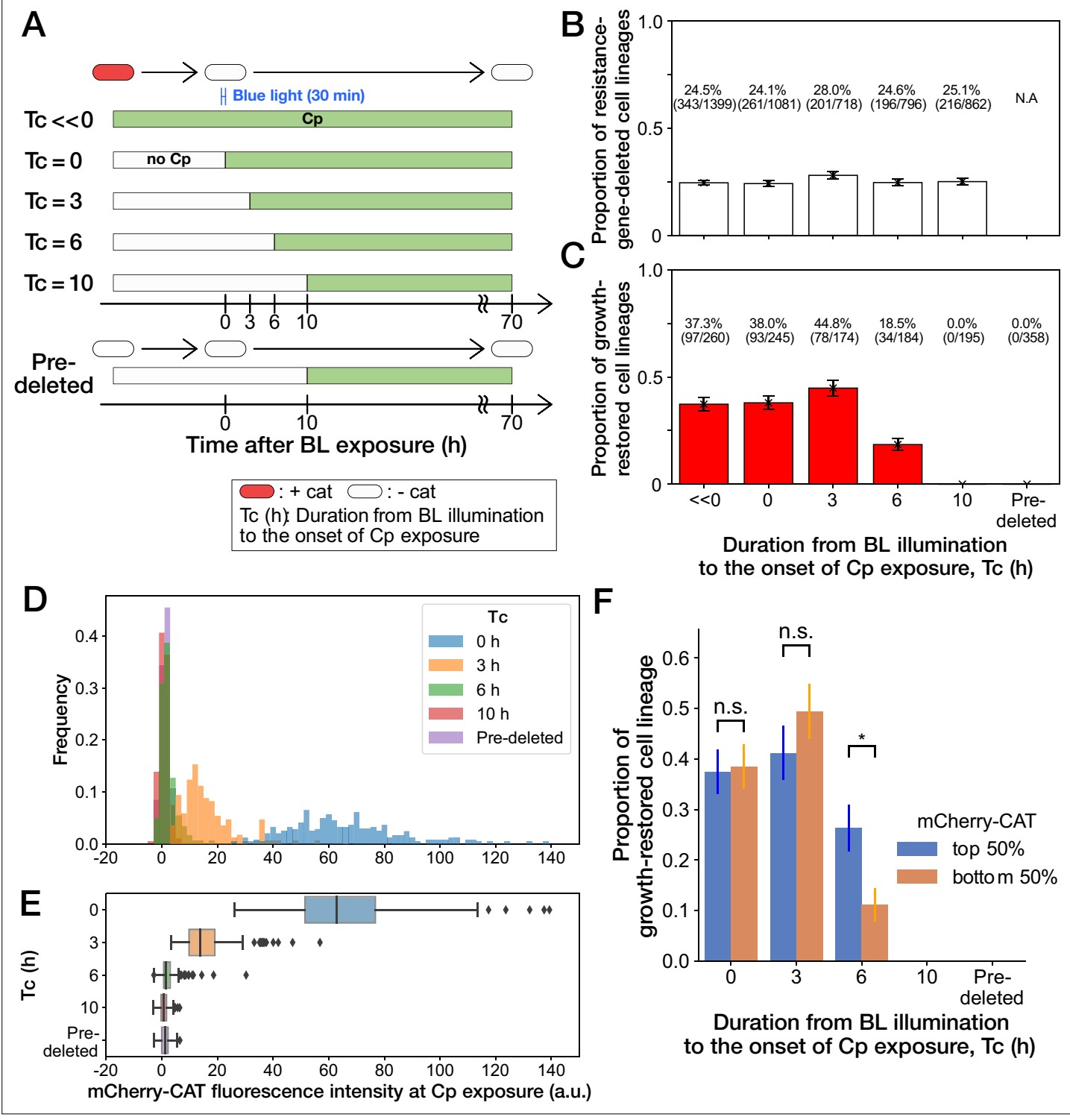

**Figure 4.** History-dependent maintenance of Cp resistance. (**A**) The schematic diagram of experiments. The duration from the end of blue-light illumination to the onset of Cp exposure ($T_c$) was varied from 0 to 10 h. $T_c \ll 0$ represents the continuous Cp-exposure condition. "Pre-deleted" denotes the results of the experiments performed with the *mcherry-cat*-deleted YK0085 strain. The non-resistant YK0085 cells were not exposed to blue light. (**B**) Fractions of *mcherry-cat*-deleted cell lineages across different $T_c$ conditions. The numbers of resistance-gene-deleted cell lineages among the total numbers of cell lineages observed during the measurements are shown above the bars. Error bars represent standard errors. (**C**) Fractions of growth-restored cell lineages among resistance-gene-deleted cell lineages. The numbers of growth-restored cell lineages among all the resistance-gene-deleted cell lineages are shown above the bars. Error bars represent standard errors. The numbers of resistance-gene-deleted cell lineages

*Figure 4 continued on next page*

*Figure 4 continued*

are different from those in B because some cell lineages were flushed away from the growth channels at the later time points of the measurements, and their fates could not be determined. (**D, E**) The distributions of the mCherry-CAT fluorescence intensities at the onset of Cp exposure across the different $T_c$ conditions represented by histograms (**D**) and box plots (**E**). (**F**) Fractions of growth-restored cell lineages and their dependence on the mCherry-CAT fluorescence at the onset of Cp exposure. Blue and orange bars indicate the fractions of growth-restored cells among cell lineages whose mCherry-CAT fluorescence intensities were higher and lower than the median, respectively. Error bars represent standard errors. The growth-restored cell lineages were not detected under the $T_c$ = 10 h and "Pre-deleted" conditions. Fractional differences between the top 50% and the bottom 50% of cell lineages were statistically significant only for the $T_c$ = 6 h condition (p = 0.86 for $T_c$ = 0 hr; p = 0.28 for $T_c$ = 3 hr; and p = 8.6 × 10$^{-3}$ for $T_c$ = 6 hr, two proportional z-test).

illumination to the onset of Cp exposure ($T_c$) between 0 h to 10 h (**Figure 4A**). The proportions of cells that lost the *mcherry-cat* gene in response to blue-light illumination almost remained unchanged among all conditions, ranging from 24% to 28% (**Figure 4B**). However, the proportions of growth-restored cell lineages among resistance-gene-deleted cells strongly depended on $T_c$ (**Figure 4C**). When Cp exposure was initiated immediately after the blue-light illumination ($T_c$ = 0 hr) or after 3 h ($T_c$ = 3 hr), the proportions of growth-restored cells were nearly equivalent to those observed under continuous Cp exposure conditions (**Figure 4C**). In contrast, the proportions of growth-restored cells were reduced to 18.5% (34/184) when $T_c$ = 6 hr, and we did not detect any growth restoration when $T_c$ = 10 hr (0/195; **Figure 4C**). The almost equivalent frequencies of growth-restored cell lineages under the $T_c$ = 0 hr and $T_c$ = 3 hr conditions and those observed under continuous exposure exclude the possibility that growth restoration requires prior exposure to Cp before resistance gene deletion.

We next conjectured that a low amount of mCherry-CAT proteins is required at the onset of Cp exposure to withstand and restore growth without the resistance gene. In fact, *mcherry-cat* gene deletion before Cp exposure led to the dilution of mCherry-CAT proteins by growth by the time of Cp exposure (**Figure 4D and E**).

To examine whether the mCherry-CAT concentration in individual cells affects growth restoration, we divided the resistance-gene-deleted cell lineages into two groups under each condition (top 50% and bottom 50%) based on their mCherry-CAT fluorescence intensities at the onset of Cp exposure. While we found no significant differences in the $T_c$ = 0 and 3 hr conditions, the top 50% group produced 2.4-fold more growth-restored cells than the bottom 50% group in the $T_c$ = 6 hr condition (**Figure 4F**). In the $T_c$ = 10 hr condition, we could barely detect the fluorescence signals from any cells (**Figure 4D and E**), and no cells showed growth restoration as mentioned above (**Figure 4F**). These results suggest that a low level of residual mCherry-CAT proteins is required at the onset of Cp exposure, but high amounts do not necessarily ensure the growth restoration in all resistance-gene-deleted cells. Moreover, 40% could be considered as the maximum frequencies at which cells can restore growth without the resistance gene.

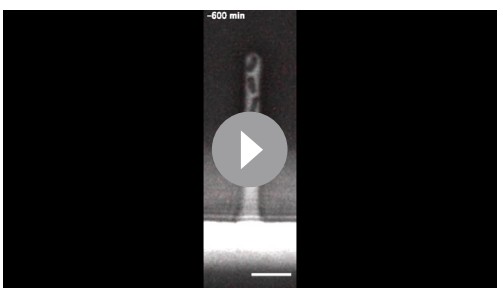

**Video 4.** No growth restoration of YK0085 strain under Cp exposure. YK0085 cells were cultured in the mother machine flowing the M9 minimal medium and exposed to 15 µg/mL of Cp at *t* = 0 min (marked with ' + Cp'). The Cp exposure caused growth arrest, and, unlike the YK0083 cells, the cell did not restore growth. We found no growth-restored cell lineages in this experimental condition. Scale bar, 5 µm.

https://elifesciences.org/articles/74486/figures#video4

## Growth restoration accompanies the recovery of stoichiometric balance of ribosomal subunits

The results above demonstrated that the residual mCherry-CAT proteins are important for initiating the transition to growth restoration without the resistance gene. However, the residual proteins cannot directly support the growth in later generations under Cp exposure as their levels are eventually diluted with growth. Indeed, the amounts of mCherry-CAT proteins during growth restoration were below the detection limit (**Figure 3F and H**). Therefore, we speculated that the other physiological changes were responsible for growth restoration.

Because Cp targets the 50 S ribosomal subunit (**Pongs, 1979**), it is plausible that ribosomal states were modulated in the duration beginning

from resistance gene deletion to growth restoration. Hence, we constructed an *E. coli* strain YK0136 that expressed fluorescently-tagged dual ribosomal reporters, RplS-mCherry and RpsB-mVenus (*Figure 5A*; *Nikolay et al., 2014*). RplS and RpsB are ribosomal proteins in the 50 S and 30 S subunits, respectively. Their fluorescent reporters were utilized to probe the ribosomal subunit balance in living cells (*Nikolay et al., 2014*). YK0136 also harbors the *cat* resistance gene (not tagged with *mcherry*) between the upstream and downstream *loxP* sequences on the chromosome, and the PA-Cre recombination system was expressed via the plasmid (*Figure 5A*). We confirmed that YK0136 showed MIC values almost equivalent to those of the YK0083 strain (*Figure 2* and *Figure 5—figure supplement 1*). Furthermore, 30-min blue-light illuminations provoked *cat*-gene deletion in 28.6% (452/1587) of YK0136 cells, which was comparable to the frequency of *mcherry-cat*-gene deletion in YK0083 (*Figure 4B*).

We conducted time-lapse measurements of YK0136 cells in the mother machine and induced the deletion of the resistance gene via blue-light illumination under continuous exposure to 15 μg/mL of Cp (*Video 5*). Since previous experiments showed that the resistance gene deletion decelerates cellular growth significantly, the deletion of the *cat* gene in each cell lineage was evaluated based on the distinctive declines of cellular elongation rates after the blue-light illumination. We validated this criterion by hierarchical time-series clustering applied to the growth of YK0083 cells and mCherry-CAT fluorescence transitions (*Figure 5—figure supplement 2* and Materials and methods).

As observed previously, 45.9% (158/344) of the *cat*-deleted YK0136 cells gradually restored their growth after initial growth suppression (*Figure 5B*, *Figure 5—figure supplement 3*, and *Video 5*). Furthermore, such growth-restored cell lineages were not observed (0/1138) when YK0138 cells were exposed to 15 μg/mL of Cp; the YK0138 strain was constructed from YK0136 cells by deleting the *cat* gene beforehand in batch culture via blue-light illumination (*Figure 5B* and *Video 6*).

The transitions of fluorescence intensities reveal that *cat*-gene deletion under Cp exposure increased the expression levels of both RplS-mCherry and RpsB-mVenus (*Figure 5C and D*). The relative changes triggered by deletion were more significant for RplS-mCherry than those for RpsB-mVenus (*Figure 5C and D*). Consequently, the ribosomal subunit balance probed by the RplS-mCherry/RpsB-mVenus ratio was disrupted, and the ratio increased approximately 3-fold (*Figure 5E and F*). The initial disruption kinetics was similar between growth-restored and growth-halted resistance-gene-deleted cell lineages (*Figure 5E*). However, the ratio of growth-restored cell lineages gradually returned to the original level, whereas the ratio of growth-halted cell lineages remained disrupted (*Figure 5E and F*). The difference in the ratio between growth-restored and growth-halted cell lineages became evident approximately 37 hr after resistance gene deletion (*Figure 5—figure supplement 4*). These results suggest that the regain of the ribosomal subunit balance under Cp exposure is correlated with growth restoration.

Analysis of the correlations between the pre-illumination phenotypic traits and post-illumination cellular fates reveals that elongation rates, fluorescence intensities of RplS-mCherry and RpsB-mVenus, and the fluorescence ratio observed before blue-light illumination did not strongly affect the likelihood of gene deletion and the determination of growth-halting and growth-restoration fates (*Figure 5—figure supplement 5*).

When non-resistant YK0138 cells were exposed to 15 μg/mL of Cp, the expression levels of RplS-mCherry increased initially following kinetics similar to those of the YK0136 cells observed after *cat*-gene deletion (*Figure 5C*). In contrast, the expression levels of RpsB-mVenus decreased in response to Cp exposure (*Figure 5D*), as previously reported for a wildtype *E. coli* strain exposed to Cp (*Siibak et al., 2011*). The initial decrease of RpsB-mVenus expression led to a more rapid disruption of the RplS-mCherry/RpsB-mVenus ratio in the susceptible YK0138 cells (*Figure 5E and F*). Consistently, the decline in the elongation rates of the YK0138 cells was faster than those of the YK0136 cells (*Figure 5B*).

We further examined the relationship between the RplS-mCherry/RpsB-mVenus ratio and elongation rates and observed a negative correlation (Spearman $\rho$ = –0.58, 95% confidence interval [-0.62,–0.53]; *Figure 6A*). The ratio was maintained within a narrow range (0.84–1.18, 95% interval) before blue-light illumination (*Figures 5F and 6A*). The disruption of ribosomal subunit balance exceeding this range was linked to growth suppression (*Figure 6A*). The relationships between the ratio and elongation rates were similar among growth-restored resistance-gene-deleted cell lineages, growth-halted resistance-gene-deleted cell lineages, and pre-deleted cell lineages (*Figure 6A*). However, the

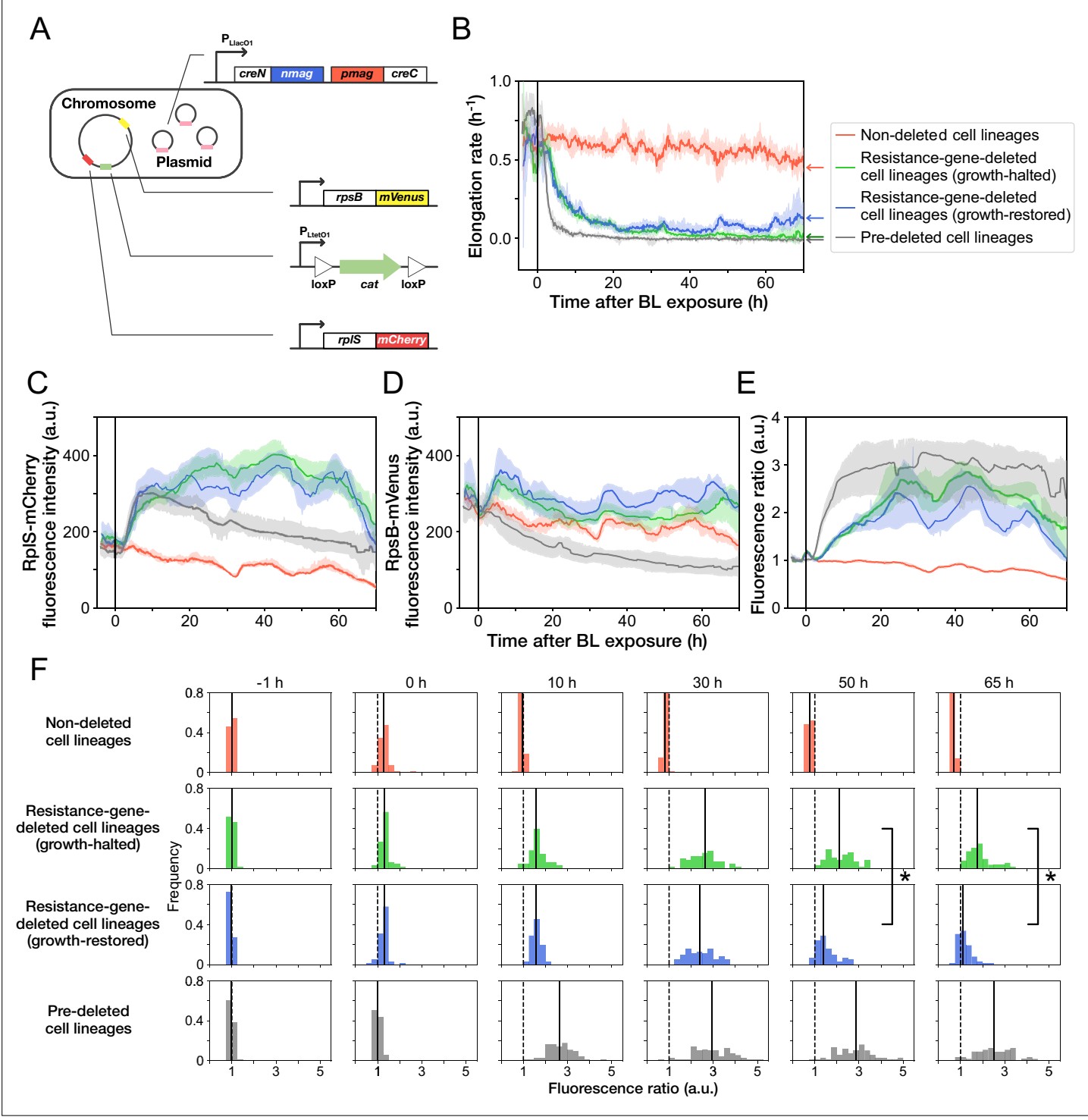

**Figure 5.** Disruption and restoration of ribosomal proteins' stoichiometry. (**A**) Schematic diagram of the ribosome reporter strain, YK0136. Fluorescently tagged ribosomal protein genes (*rplS* and *rpsB*) are expressed from the native loci on the chromosomes. Additionally, the photo-removable *cat* gene was integrated into the *intC* locus on the chromosome. The PA-Cre fragments were expressed via low-copy plasmids. (**B**) Transitions of elongation rates. Lines represent the medians of elongation rates at each time point, and shaded areas show the 95% error ranges of the medians estimated by resampling the cell lineages 1000 times. Red represents non-deleted cell lineages. Green shows growth-halted resistance-gene-deleted cell lineages. Blue represents growth-restored resistance-gene-deleted cell lineages. Gray represents pre-deleted cell lineages. Color correspondence remains the same in the following figures. The end of blue-light illumination was set to 0 hr on the horizontal axis. (**C–E**) Transitions of RplS-mCherry fluorescence intensities (**C**), RpsB-mVenus fluorescence intensities (**D**), and the fluorescence ratio of RplS-mCherry/RpsB-mVenus (**E**). The fluorescence intensities of

*Figure 5 continued on next page*

*Figure 5 continued*

RplS-mCherry and RpsB-mVenus were normalized by the intensity observed before blue light illumination to calculate the ratio in E. (**F**) Transitions of the RplS-mCherry/RpsB-mVenus fluorescence ratio distributions. The vertical dashed lines represent the position of ratio = 1; the solid lines represent the medians of the distributions. Significant differences in fluorescence ratio were detected between the growth-restored and growth-halted cell lineages at 50 hr and 65 hr (p = $5.6 \times 10^{-10}$ at 50 hr, p = $1.9 \times 10^{-13}$ at 65 hr, Mann-Whitney U test).

The online version of this article includes the following figure supplement(s) for figure 5:

**Figure supplement 1.** MIC tests of the ribosome reporter strains.

**Figure supplement 2.** Time-course transitions of elongation rates are sufficient for classifying resistance-gene-deleted and non-deleted cell lineages.

**Figure supplement 3.** Transitions of generation time and RplS-mCherry/RpsB-mVenus fluorescence ratio observed in ribosome reporter strain shown in generation.

**Figure supplement 4.** Transition of p-value from the Mann-Whitney U test applied for the RplS-mCherry/RpsB-mVenus fluorescence ratio.

**Figure supplement 5.** Relationships between cellular phenotypes before blue-light illumination and cellular fates.

distribution of the points was shifted toward the original fluorescence ratio for the growth-restored resistance-gene-deleted cell lineages (*Figure 6B–D*). These results suggest that the ribosomal subunit balance is correlated with growth under Cp exposure and that restoration of the subunit balance might help cells to recover growth in the absence of the resistance gene.

## Discussion

Uncovering the genotype-phenotype correspondences in various biological contexts is the groundwork for genetics. However, the correspondences are usually investigated based on terminal phenotypes, which are often manifested after significant lags following genotypic changes. Consequently, our understanding of the dependence of terminal phenotypes on historical conditions and the heterogeneity of phenotypic consequences among individual cells remains limited. In this study, we demonstrated that *E. coli* cells could adapt even to a lethal genetic modification, that is the deletion of *cat* resistance gene under exposure to chloramphenicol (*Figure 3*). Importantly, such adaptation was not observed when an identical genetic modification was introduced long before Cp exposure (*Figure 4*). Therefore, whether cells could gain physiological resistance in the absence of a resistance gene depends on the timing of resistance gene deletion and environmental histories that the cells experienced.

The analyses revealed that the *cat* gene deletion under Cp exposure disrupted the stoichiometric balance of ribosomal proteins (RplS and

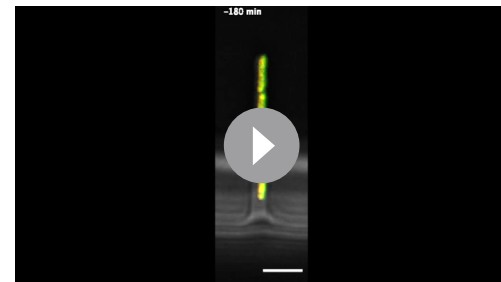

**Video 5.** Recovery of the balance of ribosomal proteins alongside growth restoration. YK0136 cells were cultured in the mother machine flowing the M9 minimal medium containing 15 µg/mL of Cp. Blue light was illuminated from *t* = –30 min to 0 min (marked with ' + BL'). The merged images of phase contrast (grayscale), RplS-mCherry fluorescence (red), and RpsB-mVenus fluorescence (green) channels are shown. The growth of the cell markedly slowed down after blue-light illumination, indicating the loss of the *cat* gene. The cell became red in response to blue-light illumination, indicating that the relative expression level of RplS-mCherry became higher than that of RpsB-mVenus. However, the balance was restored alongside growth restoration. Scale bar, 5 µm.

https://elifesciences.org/articles/74486/figures#video5

**Video 6.** No recovery of the balance of ribosomal proteins in the YK0138 strain. YK0138 cells were cultured in the mother machine flowing the M9 minimal medium and exposed to 15 µg/mL of Cp at *t* = 0 min (marked with ' + Cp'). The Cp exposure caused growth arrest and disruption of ribosomal proteins' expression balance. No restoration of growth and ribosomal balance was observed. Scale bar, 5 µm.

https://elifesciences.org/articles/74486/figures#video6

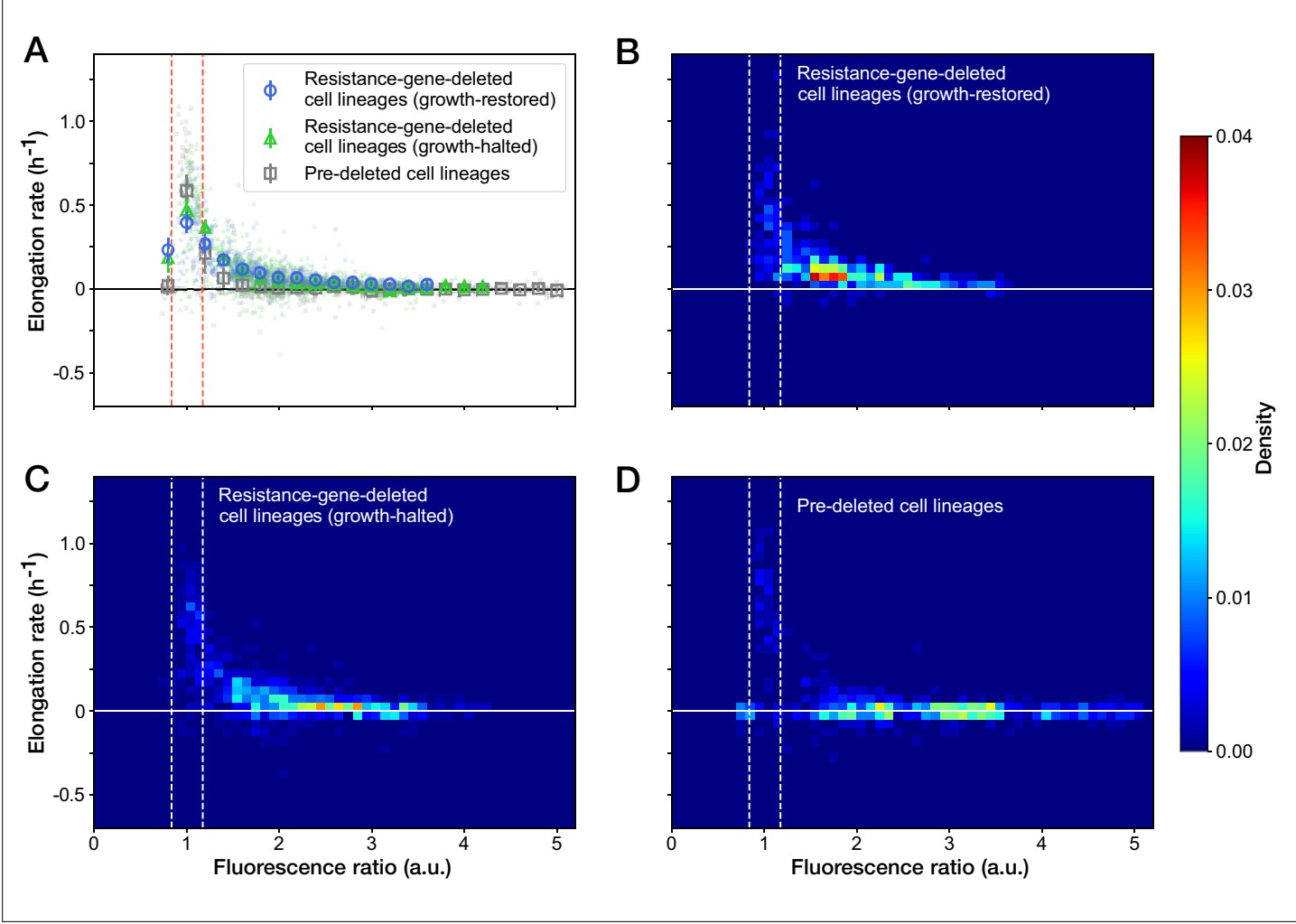

**Figure 6.** Relationship between the RplS-mCherry/RpsB-mVenus fluorescence ratio and elongation rate under Cp exposure. (**A**) Scatter plot of the RplS-mCherry/RpsB-mVenus fluorescence ratio and elongation rate. Small points represent the relations between the fluorescence ratio and elongation rates averaged over the two-hour periods at the different time points on the single-cell lineages. The larger open point represents the mean elongation rate in each bin of fluorescence ratio (bin width, 0.2 a.u.). Error bars represent the 1.96× standard error ranges. Blue represents growth-restored resistance-gene-deleted cell lineages. Green represents growth-halted resistance-gene-deleted cell lineages. Gray represents pre-deleted cell lineages. Dashed lines indicate the mean ± 1.96× standard error range of fluorescence ratio of non-deleted cell lineages before blue-light illumination. (**B–D**) Density plots of the points for the relationships between fluorescence ratio and elongation rate. (**B**) Growth-restored resistance-gene-deleted cell lineages. (**C**) Growth-halted resistance-gene-deleted cell lineages. (**D**) Pre-deleted cell lineages. Color represents the density of the points for each type of cell lineages shown in (**A**). Bin size was 0.05 hr$^{-1}$ for elongation rate and 0.1 a.u. for fluorescence ratio.

RspB) and that the balance was gradually recovered alongside growth restoration (**Figure 5**). The elongation rate and the stoichiometric balance of ribosomal proteins were strongly correlated (**Figure 6**). Furthermore, the balance was tightly regulated within a narrow range in the fast-growing non-deleted cells (**Figure 5F**). Therefore, the recovery of ribosomal stoichiometric balance might contribute to restoring growth against the deletion of the resistance gene.

It still remains elusive how individual cells recovered the ribosomal stoichiometric balance under continuous Cp exposure. A plausible scenario might be that the multi-layered and interlinked feedback regulations on ribosomal components entailed stoichiometry readjustments (**Yamagishi and Nomura, 1988**; **Nomura et al., 1984**; **Lindahl and Zengel, 1986**; **Keener and Nomura, 1996**; **Kaczanowska and Rydén-Aulin, 2007**). Many ribosomal proteins bind to their own mRNAs, in addition to the target ribosomal RNAs (**Nomura et al., 1984**; **Lindahl and Zengel, 1986**; **Keener and Nomura, 1996**; **Kaczanowska and Rydén-Aulin, 2007**). These regulatory fractions of ribosomal proteins bind to their mRNAs and inhibit their own translation and that of other genes in the same operons. The competition

among rRNA and mRNA for these regulatory ribosomal protein components is considered to regulate the stoichiometry of ribosomal components (*Nomura et al., 1984*; *Lindahl and Zengel, 1986*; *Keener and Nomura, 1996*; *Kaczanowska and Rydén-Aulin, 2007*). Therefore, it is plausible that these innate homeostatic mechanisms of ribosomes might have been exploited by cells to restore growth. It is also important to note that the operons for ribosomal proteins comprise genes encoding RNA polymerase subunits, translation initiation/elongation factors, DNA primase and protein transmembrane transporters (*Lindahl and Zengel, 1986*; *Keener and Nomura, 1996*). Hence, the feedback regulations in ribosomes could mediate changes in the global expression profiles, which might have helped the cells find growth-permissive states in the presence of Cp.

The necessity for a small amount of residual resistant proteins at the onset of Cp exposure (*Figure 4*) might suggest that experiencing such barely-tolerable intracellular states is crucial for gaining physiological resistance. The importance of near-critical states for survival might be related to the phenomenon that pre-treatment with a sublethal concentration of antibiotics increases the fraction of persister cells in response to the subsequent treatments with lethal antibiotic concentrations (*Dörr et al., 2009*; *Johnson and Levin, 2013*). We also remark that the physiological adaptation to the antibiotic stress caused by resistance gene deletion might be related to *adaptive resistance*. Adaptive resistance is a phenomenon in which bacterial populations progressively enhance their resistance levels (MICs) when subject to gradual increases of antibiotics (*George and Levy, 1983*; *Adam et al., 2008*; *Sánchez-Romero and Casadesús, 2014*; *Motta et al., 2015*). Adaptive resistance is achieved physiologically rather than genetically since the enhanced resistance is lost when the antibiotic is removed. Therefore, it is plausible that growth-restoration against resistance gene deletion might be realized by some mechanisms common to adaptive resistance. The fact that no cells recovered growth when the resistance gene was deleted at a high Cp concentration (30 μg/mL) also suggests a link to adaptive resistance; the speed of antibiotic stress intensification must be sufficiently slow for the cells to gain resistance physiologically. However, we also remark that it is nontrivial whether bacterial cells respond to the intensification of antibiotic stress caused by internal resistance-gene deletion similarly to the stress caused by gradual increases of external antibiotic concentrations. Although our results might support the hypothesis that physiological adaptation to genetic perturbations that sensitize cells to some antibiotics is feasible when bacterial cells exhibit adaptive resistance to those drugs, more careful examinations under diverse experimental conditions should be needed to understand the identity of these internally and externally intensified antibiotic stress. Additionally, we note that adaptive resistance has been investigated at the population level, not at the single-cell level, in the previous studies (*George and Levy, 1983*; *Adam et al., 2008*; *Sánchez-Romero and Casadesús, 2014*). Consequently, it remains elusive whether only a small fraction of cell lineages in the populations acquire resistance and dominate the populations by selection, or most cells can progressively enhance resistance in adaptive resistance. In this study, using the mother machine microfluidic device, we were able to quantify the fractions of cell lineages that restore growth against resistance-gene deletion and the changes of the fractions depending on the historical conditions. Such single-cell-level quantitative information would be valuable for unraveling the windows of phenotypic space and the ranges of conditions permitting physiological adaptation to antibiotics.

Though experimental evidence of history-dependent physiological adaptation to lethal genetic perturbations is still limited, the experimental strategy combining optogenetic recombination and single-cell lineage tracking is applicable for the modification of other genes or other cell types, including mammalian cells. Such approaches might offer new insights into the emergence of drug-resistant bacterial and cancer cells. We remark that growth-restoration of resistance-gene-deleted cells must have been hardly recognized in batch-culture experiments because the fractions of these slow-growing cells decrease in time by selection within cell populations (*Figure 3—figure supplement 2B*). Therefore, our study also confirms the advantage of microfluidic time-lapse microscopy for unraveling long-term adaptation phenomena that occur in slow-growing cell lineages. Gaining a more comprehensive picture of resistance evolution might help us design new drug treatment strategies to counter or control the emergence and spread of resistance.

## Materials and methods

### Key resources table

| Reagent type (species) or resource | Designation | Source or reference | Identifiers | Additional information |
|---|---|---|---|---|
| Strain, strain background (*Escherichia coli*) | F3 | [31] | | W3110 Δ*fliC*::FRT Δ*fimA*::FRT Δ*flu*::FRT |
| Strain, strain background (*Escherichia coli*) | YK0083 | This study | | F3 *intC*::P$_{LtetO1}$-loxP-RBS4-*mcherry-cat*-loxP-FRT/pYK022 |
| Strain, strain background (*Escherichia coli*) | MUS6 | This study | | F3 *rplS-mcherry*-FRT *rpsB-mvenus*-FRT |
| Strain, strain background (*Escherichia coli*) | YK0136 | This study | | MUS6 *intC*::P$_{LtetO1}$-loxP-RBS4-*cat*-loxP-FRT/pYK022 |
| Recombinant DNA reagent | *pmag-creC* | [12] | | gifted from Sato lab. |
| Recombinant DNA reagent | *nmag-creN* | [12] | | gifted from Sato lab. |
| Recombinant DNA reagent | pYK022 | This study | | plasmid expressing PA-Cre genes. |
| Sequence-based reagent | intC_check_F | This study | PCR primer for intC check | GATCGATACTTG CTGTGGTTGATG |
| Sequence-based reagent | intC_check_R2 | This study | PCR primer for intC check | CCTCTTAGTTAAATG GATATAACGAGCCCC |
| Sequence-based reagent | YKp0077 | This study | PCR primer for cat check | CACCGTTGATA TATCCCAATGGC |
| Sequence-based reagent | YKp0115 | This study | PCR primer for cat check | CACTCATCG CAGTACTGTTG |
| Commercial assay or kit | Wizard SVGel and PCRClean-Up System | Promega | Cat.# A9281 | |
| Commercial assay or kit | PureYieldPlasmid MiniprepSystem | Promega | Cat.# A1223 | |
| Commercial assay or kit | WizardGenomic DNAPurification Kit | Promega | Cat.# A1120 | |
| Chemical compound, drug | chloramphenicol | Wako | 030-19452 | |
| Chemical compound, drug | isopropyl-β-D-thiogalactopyranoside | Wako | 094-05144 | |
| Chemical compound, drug | Bovine Serum Albumin | Sigma-Aldrich | Cat.# A6003 | |
| Chemical compound, drug | polydimethylsiloxane | Dow Corning | SYLGARD 184 | |
| Software, algorithm | FastDTW | [32] | | |
| Software, algorithm | HyperStackReg | [33] | | |

## Bacterial strains, plasmids construction, and culture conditions

*E. coli* strains, plasmids, and primers are listed in *Table 2*, *Table 3* and *Table 4*, respectively. For strain constructions, cells were grown in Luria Bertani (LB) broth (Difco) at 37 °C unless stated otherwise. Plasmids were constructed using DNA ligase (T4 DNA Ligase, Takara) or by ExoIII cloning (*Li and Evans, 1997*) and introduced into the *E. coli* strain JM109 for cloning and stocking. The *cat* or *mcherry-cat* gene was introduced into the *intC* locus of F3 (W3110Δ*fliC*::FRTΔ*fimA*::FRTΔ*flu*::FRT)

**Table 2.** Strain list used in this study.

| | mCherry-CAT deletion | |
|---|---|---|
| **Name** | **Genotype** | **Source** |
| F3 | W3110 Δ*fliC*::FRT Δ*fimA*::FRT Δ*flu*::FRT | *Hashimoto et al., 2016* |
| YK0080 | F3 *intC*::P$_{LtetO1}$-loxP-RBS4-*mcherry-cat*-loxP-FRT | This study |
| YK0083 | F3 *intC*::P$_{LtetO1}$-loxP-RBS4-*mcherry-cat*-loxP-FRT/pYK022 | This study |
| YK0085 | F3 *intC*::P$_{LtetO1}$-FRT/pYK022 | This study |
| | **Ribosome reporter** | |
| **Name** | **Genotype** | **Source** |
| MUS3 | BW25113 *rplS-mcherry*-FRT-*kan*-FRT | This study |
| MUS13 | BW25113 *rpsB-mvenus*-FRT-*kan*-FRT | This study |
| MUS5 | F3 *rplS-mcherry*-FRT | This study |
| MUS6 | F3 *rplS-mcherry*-FRT *rpsB-mvenus*-FRT | This study |
| YK0134 | MUS6 *intC*::P$_{LtetO1}$P-loxP-RBS4-*cat*-loxP-FRT | This study |
| YK0136 | MUS6 *intC*::P$_{LtetO1}$-loxP-RBS4-*cat*-loxP-FRT/pYK022 | This study |
| YK0138 | MUS6 *intC*::P$_{LtetO1}$-loxP-FRT/pYK022 | This study |

**Table 3.** Plasmid list used in this study.

| Name | Backbone | Gene | Source |
|------|----------|------|--------|
| pKD46 | | | CGSC |
| pCP20 | | | CGSC |
| | pcDNA3.1 | pCMV-lox2272-loxP-inversed *mvenus*-lox2272-loxP | Sato lab |
| | pcDNA3.1 | pCMV-rfp-*pmag-creC* | Sato lab |
| | pcDNA3.1 | pCMV-rfp-*nmag-creN* | Sato lab |
| pTmCherryK3 | pMW118 | $P_{LtetO1}$-RBS3-*mcherry*-FLP-*kan*-FLP | Wakamoto lab |
| pLVK4 | pMW118 | $P_{LlacO1}$-RBS4-*mvenus*-FLP-*kan*-FLP | Wakamoto lab |
| pTVCK4 | pMW118 | $P_{LtetO1}$-RBS4-*mvenus-cat*-FLP-*kan*-FLP | Wakamoto lab |
| pKK1 | pMW118 | $P_{LlacO1}$-rfp-*pmag-creC*-FLP-*kan*-FLP | This study |
| pKK2 | pMW118 | $P_{LlacO1}$-rfp-*nmag-creN*-FLP-*kan*-FLP | This study |
| pYK001 | pMW118 | $P_{LlacO1}$-RBS3-*pmag-creC*-FLP-*kan*-FLP | This study |
| pYK002 | pMW118 | $P_{LlacO1}$-RBS3-*nmag-creN*-FLP-*kan*-FLP | This study |
| pYK006 | pMW118 | $P_{LtetO1}$-RBS3-lox2272-*mvenus*-loxP-FLP-*kan*-FLP | This study |
| pYK007 | pMW118 | $P_{LtetO1}$-RBS3-lox2272-*mcherry*-loxP-FLP-*kan*-FLP | This study |
| pYK008 | pMW118 | $P_{LtetO1}$-RBS3-lox2272-*mcherry*-loxP-FLP-*kan*-FLP | This study |
| pYK009 | pMW118 | $P_{LtetO1}$-RBS3-lox2272-*mcherry*-loxP-FLP-*kan*-FLP | This study |
| pYK010 | pMW118 | $P_{LtetO1}$-RBS4-lox2272-*mcherry*-loxP-FLP-*kan*-FLP | This study |
| pYK011 | pMW118 | $P_{LtetO1}$-lox2272-RBS4-*mcherry*-loxP-FLP-*kan*-FLP | This study |
| pYK016 | pMW118 | $P_{LlacO1}$-RBS3-*pmag-creC*-FLP-*kan*-FLP | This study |
| pYK017 | pMW118 | $P_{LlacO1}$-RBS3-*creN-nmag*-FLP-*kan*-FLP | This study |
| pYK018 | pMW118 | $P_{LlacO1}$-RBS3-*creN-nmag-pmag-creC*-FLP-*kan*-FLP | This study |
| pYK022 | pMW118 | $P_{LlacO1}$-RBS3-*creN-nmag-pmag-creC*-FLP-*kan*-FLP | This study |
| pYK023 | pMW118 | $P_{LtetO1}$-loxP-RBS4-*mcherry*-loxP-FLP-*kan*-FLP | This study |
| pYK028 | pMW118 | $P_{LtetO1}$-loxP-RBS4-*mcherry*-loxP-FLP-*kan*-FLP | This study |
| pYK029 | pMW118 | $P_{LtetO1}$-loxP-RBS4-*mcherry-cat*-loxP-FLP-*kan*-FLP | This study |
| pYK035 | pMW118 | $P_{LtetO1}$-loxP-RBS4-*cat*-loxP-FLP-*kan*-FLP | This study |
| pMU1 | pMW118 | $P_{LlacO1}$-RBS4-*mcherry*-FLP-*kan*-FLP | This study |
| pMU2 | pMW118 | $P_{LlacO1}$-RBS4-*mvenus*-FLP-*kan*-FLP | This study |

genome by λ-Red recombination (YK0080, YK0134) (*Datsenko and Wanner, 2000*). Fluorescently-tagged ribosome reporter genes were introduced into the *E. coli* strain BW25113 (MUS3, MUS13). These ribosome reporter genes were transferred into the F3 strain by P1 transduction (MUS5, MUS6). The colonies with intended genome integration were selected on LB plates containing kanamycin (Km, Wako). The Km resistant gene was removed by flp-FRT recombination using pCP20 plasmid (*Datsenko and Wanner, 2000*).

When constructing the resistance-gene-deleted strains (YK0085 and YK0138), YK0083 and YK0136 cells were pre-cultivated overnight in LB broth containing 50 μg/mL of ampicillin (Amp, Wako). 10 μL of overnight cultures were inoculated into 2 mL of the M9 medium containing 50 μg/mL of Amp and 0.1 mM isopropyl-β-D-thiogalactopyranoside (IPTG, Wako) in test tubes. The cell cultures were cultivated at 37 °C with shaking. After a 3-hr cultivation period for inducing the expression of PA-Cre system by IPTG, the batch cultures were illuminated by LED blue light (CCS) for 24 hr. The light intensity was adjusted to 6.8 mW, which is equivalent to the intensity in the microscopy experiments. The cell cultures were streaked on LB agar containing 100 μg/mL of Amp. The plates were incubated at 37 °C overnight. After overnight incubation, several colonies were selected. The deletion of the intended gene in these colonies was examined via PCR. The colonies with the intended gene deletion

**Table 4.** Primer list used in this study.

| PA-Cre plasmid construction | | |
|---|---|---|
| Name | Sequence | Usage |
| pLVK_pTVK_F | CAGGCATCAAATAAAACGAAAGGCTCAGTC | |
| pLVK3_pTVK3_R | GGTACCTTTCTCCTCTTTAATGTTTTCGG | |
| L3_TagRFP_F | GGACGCACTGACCGAAAACATTAAAGAG GAGAAAGGTACCATGGTGTCTAAGGGCGAAG | For constructing pKK1 and pKK2 plasmids using ExoIII cloning (**Li and Evans, 1997**). Templates: pLVK3 (pLVK_pTVK_F & pLVK3_pTVK3_R), pCMV-rfp-pmag-creC (L3_TagRFP_F & Crec_Term_R), and pCMV-rfp-nmag-creN (L3_TagRFP_F & nMagCreN-R) |
| Crec_Term_R | CCAGTCTTTCGACTGAGCCTTTCGTTTT ATTTGATGCCTGTTAGTCCCCATCTTCGAGCAG | |
| nMagCreN-R | TTCGTTTTATTTGATGCCTGTTAGTTCAGCTTGCACCAGG | |
| YKp0001 | ATGCATACTCTTTATGCCCCCGG | For constructing pYK001 and pYK002 plasmids by removing the *rfp* gene from pKK1 and pKK2. |
| YKp0002 | GGTACCTTTCTCCTCTTTAATG | |
| YKp0053 | AACAGGAAATGGTTCCCTGCTGAACC | For constucting pYK016 plasmid by changing the linker between *pmag* and *creC* on pYK001. |
| YKp0054 | GGTACCTTCTGTTTCGCACTGGAATC | |
| YKp0022 | GTGTAGGCTGGAGCTGCTTCG | |
| YKp0049 | AAGAGGAGAAAGGTACCATGACCTCTGATGAAGTCAGG | |
| YKp0052 | CATGGTACCTTTCTCC | |
| YKp0055 | CATACTCTTTATGCCCCCGGTGG | |
| YKp0056 | GCATAAAGAGTATGGGTACCGTTCAGCTTGCACCAG | For constructing pYK017 plasmid using ExoIII cloning. PCR was conducted with the template pYK002 using YKp0022 & YKp0052, YKp0049 & YKp0056 and YKp0055 & YKp0057 to change the order of *nmag* and *creN*. |
| YKp0057 | CTCCAGCCTACACTTATTCTGTTTCGCACTGGAATC | |
| YKp0058 | TTATTCTGTTTCGCACTGGAATCCC | |
| YKp0059 | GCGAAACAGAATAAAGGAGAAAGGTACCATGCATAC | For constructing pYK018 plasmid using ExoIII cloning. PCR was conducted on the template pYK016 with YKp0059 & YKp0060 and pYK017 with YKp0022 & YKp0058 to combine *pmag-creC* and *creN-nmag*. |
| YKp0060 | GCTCCAGCCTACACTTAGTCCCCATCTTCGAGCAGC | |
| YKp0006 | CAGGCATCAAATAAAACGAAAGGCTCAGTCG | For constructing pYK022 plasmid using ExoIII cloning. PCR was conducted on the template pYK001 with YKp0006 & YKp0052 and pYK018 with YKp0049 & YKp0061 to add a terminator. |
| YKp0061 | TTATTTGATGCCTGTTAGTCCCCATCTTCGAGCAGC | |

| Floxed *cat* plasmid construction | | |
|---|---|---|
| Name | Sequence | Usage |
| YKp0026 | CGTTTTATTTGATGCCTGATAACTTCGTATAGCATAC | For constructing pYK006 plasmid. Template: pCMV_lox2272_loxp_inverse-mvenus_lox2272_loxp. |
| YKp0027 | CATTAAAGAGGAGAAAGGTACCATAACTTCGTATAGGATAC | |
| YKp0029 | CTTATTAGAATTCGCCGCCATGGTGAGCAAGGGCGAG | |
| YKp0030 | CATTATACGAAGTTATCTCGAGTTATCCACGCGTGAGC | |
| YKp0031 | CTCGAGATAACTTCGTATAATG | For constructing pYK007 plasmid using ExoIII cloning. PCR was conducted on the template pTmCherryK3 with YKp0029 & YKp0030 and pYK006 with YKp0031 & YKp0032 to convert *mvenus* to *mcherry*. |
| YKp0032 | GGCGGCGAATTCTAATAAGG | |
| YKp0035 | ATAACTTCGTATAAAGTATCCTATACGAAGTTATGGTACC | For constructing pYK008 plasmid. PCR was conducted on the template pYK007 to delete spacer sequence. |
| YKp0036 | ATGGTGAGCAAGGGCGAGGAGGATAACATGGCCATCATC | |
| YKp0037 | TAACTCGAGATAACTTCGTATAATG | For constructing pYK009 plasmid with the template pYK008 to delete SacI and MluI restriction sites. |
| YKp0038 | CTTGTACAGCTCGTCCATGC | |
| YKp0039 | GAAAAAAATAACTTCGTATAGGATAC | For constructing pYK010 plasmid with the template pYK009 to change the ribosomal binding site (RBS). |
| YKp0040 | CTCCTCTTTAATGTTTTCGGTCAGTGCG | |
| YKp0041 | GAAGTTATAGGAGGAAAAAAATGGTGAGCAAGGGCGAGGAGG | |
| YKp0042 | GTATAAAGTATCCTATACGAAGTTATCT TTAATGTTTTCGGTCAGTGCG | For constructing pYK011 plasmid with the template pYK010 to transfer RBS in floxed site. |
| YKp0069 | CCGAAAACATTAAAGATAACTTCGTATAGCATACATTATACG | For constructing pYK023 with the template pYK011 (YKp0041 & YKp0069) to change lox2272 to loxP. |

*Table 4 continued on next page*

*Table 4 continued*

| PA-Cre plasmid construction | | |
|---|---|---|
| Name | Sequence | Usage |
| YKp0083 | ATACATTATACGAAGTTATCAGGCATCAAATAAAACG | For constructing pYK028 plasmid with the template pYK023 to change the direction of loxP site. |
| YKp0084 | GCTATACGAAGTTATCTCGAGTTACTTGTACAGCTC | |
| YKp0072 | CGAGCTGTACAAGGAGCTCGAGAAAAAAATCACTGG | For constructing pYK029 plasmid using ExoIII cloning. PCR was conducted on the template pYK028 (YKp0072 YKp0089) and pTVCK4 (YKp0075 & YKp0083) to insert *cat* gene. |
| YKp0075 | CTTGTACAGCTCGTCCATGCCGCCGGTGG | |
| YKp0089 | CGTATAATGTATGCTATACGAAGTTATCTCGAGTTATCCAC | |
| YKp0109 | GAGAAAAAAATCACTGG | For constructing pYK035 plasmid with the template pYK029 to remove *mcherry* gene. |
| YKp0110 | CATTTTTTTCCTCCTATAAC | |

Genome integration by $\lambda$-Red recombination

| Name | Sequence | Usage |
|---|---|---|
| intC_PtetO1_F3 | AGTTGTTAAGGTCGCTCACTCCACCTTC TCATCAAGCCAGTCCGCCCATCCCTATCAGTGATAGAGATTG | For genome integration of floxed *mcherry-cat*-FRT-*kan*-FRT or *cat*-FRT-*kan*-FRT at *intC* site by $\lambda$-Red recombination. PCR was conducted on the template pYK029 or pYK035. |
| intC_R | CCGTAGATTTACAGTTCGTCATGGTTCG CTTCAGATCGTTGACAGCCGCAATTCCGGGGATCCGTCGACC | |
| oMU94 | ACCTGCGTGAGCGTACTGGTAAGGCTGC TCGTATCAAAGAGCGTCTTAACGTGAGCAAGGGCGAGGA | |
| oMU107 | GCCAGCCAATTGGCCAGCCCTTCTTAAC AGGATGTCGCTTAAGCGAAATCTTGTGTAGGCTGGAGCTGCT | For integrating *mcherry*-FRT-*kan*-FRT fragment at *rplS* site. Template: pMU1. |
| oMU92 | GTTCTCAGGATCTGGCTTCCCAGGCGG AAGAAAGCTTCGTAGAAGCTGAGGTGAGCAAGGGCGAGGA | |
| oMU108 | TTGCCGCCTTTCTGCAACTCGAACTATT TTGGGGGGAGTTATCAAGCCTTATTGTGTAGGCTGGAGCTGCT | For integrating *mvenus*-FRT-*kan*-FRT fragment at *rpsB* site. Template: pMU2. |

Primers for sequence check

| Name | Sequence | Usage |
|---|---|---|
| M13F | CAGGAAACAGCTATGAC | Sequence check for the pMW118 derivative plasmids. |
| pM1_pVT_seq_primer1 | GGCACCCCAGGCTTTAC | |
| intC_check_F | GATCGATACTTGCTGTGGTTGATG | Sequence check for the *intC* site. |
| intC_check_R2 | CCTCTTAGTTAAATGGATATAACGAGCCCC | |
| YKp0077 | CACCGTTGATATATCCCAATGGC | Sequence check for the *cat* gene. |
| YKp0115 | CACTCATCGCAGTACTGTTG | |
| oMU96 | TCCAGACTCACTCTCCGGTAGT | Sequence check for the *rplS* site. For the BW25113 derivative strains oMU96 & oMU97; For the W3110 derivative strains oMU96 & oMU110 |
| oMU97 | ATAGCCAGTAACAAGACCGCCC | |
| oMU110 | GACAAATTCCACGCAGCAATCTCAC | |
| oMU26 | AAGCAAACAACCTGGGTATTCCGGT | Sequence check for the *rpsB* site |
| oMU28 | CTCGCTCATCCCGGTCACTTACTGA | |

were cultured in LB medium containing 50 µg/mL of Amp at 37 °C. These cell cultures were stored at –80 °C as glycerol stocks.

The genomes of the constructed strains and plasmids were purified using the Promega Genomic DNA Purification kit and the Promega Plasmid Miniprep System, respectively, and the DNA sequence of the modified locus was amplified by Prime STAR (Takara). The sequence was examined by Sanger sequencing using a commercial service (FASMAC).

In microscopy experiments, we used M9 minimal medium, which consisted of M9 salt (Difco), 2 mM MgSO$_4$ (Wako), 0.1 mM CaCl$_2$ (Wako), 0.2% (w/v) glucose (Wako), and 0.2%(v/v) MEM amino acid (50 x) solution (Sigma). When necessary, antibiotics and chemicals were added at the following final

concentrations unless otherwise noted: Amp 50 µg/mL, Km 20 µg/mL, chloramphenicol (Cp, Wako) 15 µg/mL, and IPTG 0.1 mM.

## Calculation of the fraction of resistance-gene-deleted cells in batch culture

YK0083 cells were pre-cultivated in LB broth containing 50 µg/mL of Amp overnight. 100 µL of the overnight culture was spun at 21,500 ×g for 1 min by a centrifuge (CT15E, himac, Hitachi) and resuspended in 1 mL of the M9 minimal medium. This culture was inoculated in 2 mL of the M9 medium containing 50 µg/mL of Amp and 0.1 mM IPTG (except for the 'no IPTG' condition) in test tubes covered with aluminum foil for light protection. The starting $OD_{600}$ was adjusted to 0.001, and the cells were cultivated at 37 °C with shaking. After a 3-hr cultivation period for inducing the expression of the PA-Cre system by IPTG, the aluminum foil cover was removed, and the test tubes were exposed to blue light. The light intensity was adjusted to 6.8 mW. The illumination length was determined according to the experimental conditions (*Figure 1—figure supplement 2B*). For the conditions with an illumination duration of less than 6 hr, the test tubes were again covered with aluminum foil and incubated with shaking so that the total cultivation duration from the start of blue-light illumination was identical (6 hr) across all conditions. Under the 'no IPTG' condition, the test tubes were covered with the aluminum foil throughout the incubation period and were not exposed to blue light.

For calculating the fractions of resistance-gene-deleted cells, the cultures exposed to blue-light were diluted to $OD_{600} = 1.0 \times 10^{-6}$, and 150 µL of the diluted cultures was spread on LB agar containing 100 µg/mL of Amp. The plates were incubated at 37 °C for 18 hr. After incubation, the number of colonies was counted under ambient light or excitation light for examining mCherry fluorescence using stereomicroscope (stereomicroscope: Olympus SZ61; LED source: NIGHTSEA SFA-GR). The fraction of resistance-gene-deleted cells was calculated as the number of non-fluorescent colonies divided by the number of total colonies (*Figure 1—figure supplement 2B*). To validate this classification, we conducted colony PCR and checked whether the designed deletion was introduced (*Figure 1—figure supplement 2C*).

We also performed the resistance-gene-deletion experiments under the Cp exposure (*Figure 3—figure supplement 2A*). YK0083 cells were pre-cultivated in LB broth containing 50 µg/mL of Amp overnight. 100 µL of the overnight culture was spun at 21,500 ×g for 1 min by a centrifuge and resuspended in 1 mL of the M9 minimal medium. This culture was inoculated in 2 mL of the M9 medium containing 50 µg/mL of Amp, 15 µg/mL of Cp and 0.1 mM IPTG in test tubes covered with aluminum foil for light protection. The starting $OD_{600}$ was adjusted to 0.001, and the cells were cultivated at 37 °C with shaking. After a 3 hr cultivation period for inducing the expression of the PA-Cre system by IPTG, the aluminum foil cover was removed, and the test tubes were exposed to blue light for 30 min. The test tubes were again covered with aluminum foil and incubated with shaking for 2 hr. The cultures exposed to blue-light were diluted to $OD_{600} = 1.0 \times 10^{-6}$, and 150 µL of the diluted cultures was spread on LB agar containing 100 µg/mL of Amp. The plates were incubated at 37 °C overnight. After incubation, we examined the fluorescence of mCherry under the excitation light. The correspondence between the loss of mCherry fluorescence and the absence of *cat* resistance gene was verified by colony PCR (*Figure 3—figure supplement 2A*).

## Sample preparation and experimental procedures for competition assay

YK0083 cells were pre-cultivated in LB broth containing 50 µg/mL of Amp overnight. 100 µL of the overnight culture was spun at 21,500 ×g for 1 min by a centrifuge and resuspended in 1 mL of the M9 minimal medium. This culture was inoculated in 2 mL of the M9 medium containing 50 µg/mL of Amp, 15 µg/mL of Cp and 0.1 mM IPTG in test tubes covered with aluminum foil for light protection. The starting $OD_{600}$ was adjusted to 0.001, and the cells were cultivated at 37 °C with shaking. After a 3-hr cultivation period for inducing the expression of the PA-Cre system by IPTG, the aluminum foil cover was removed, and the test tubes were exposed to blue light for 30 min. The test tubes were again covered with aluminum foil and incubated with shaking. The cultivation length after blue light illumination was determined according to the experimental conditions (*Figure 3—figure supplement 2B*). After the cultivation, the cultures were diluted to $OD_{600} = 1.0 \times 10^{-6}$, and 100 µL of the diluted cultures was spread on LB agar containing 100 µg/mL of Amp. The plates were incubated at 37 °C for

18 hr. After incubation, the number of colonies was counted under ambient light or excitation light for examining mCherry fluorescence using stereomicroscope. The fraction of resistance-gene-deleted cells was calculated as the number of non-fluorescent colonies divided by the number of total colonies (*Figure 3—figure supplement 2B*).

## Sample preparation and experimental procedures for MIC tests

Cells stored as a glycerol stock were inoculated in the LB broth (*Figure 2* and *Figure 5—figure supplement 1*) or the M9 medium (*Figure 3—figure supplement 3C*) containing 50 µg/mL of Amp. The cells were cultured at 37 °C with shaking overnight. 100 µL of overnight culture was centrifuged at 21,500×g for 3 min. The supernatant was discarded, and the cells were resuspended in 1.0 mL of the M9 medium. This culture was diluted to the cell density corresponding to $OD_{600}$ = 0.01 in the M9 medium containing 50 µg/mL of Amp and incubated with shaking for 3–4 hr at 37 °C. The cell culture was again diluted to $OD_{600}$ = 0.001 in fresh M9 media containing different concentrations of Cp (*Figure 2*, *Figure 3—figure supplement 3C* and *Figure 5—figure supplement 1*). After 23 hr of incubation, the $OD_{600}$ of these cultures was measured with a spectrometer (UV-1800, Shimadzu) in *Figure 2* and *Figure 5—figure supplement 1* or with a multi-mode microplate reader (FilterMax F5, Molecular Devices) in *Figure 3—figure supplement 3C*.

## Microfabrication of mother machine microfluidic device

We created two chromium photomasks, one for the main trench and the other for the observation channels, by laser drawing (DDB-201-TW, Neoark) on mask blanks (CBL4006Du-AZP, Clean Surface Technology). The photoresist on mask blanks was developed in NMD-3 (Tokyo Ohka Kogyo), and the uncovered parts of the chromium layer were removed by MPM-E30 (DNP Fine Chemicals). The remaining photoresist was removed by acetone. The photomasks were rinsed in MilliQ water and air-dried.

We created the mold for the mother machine on a silicon wafer (ID447, $\phi$=76.2 mm, University Wafer). First, we spin-coated SU8-2 (MicroChem) on the wafer with the target height of 1.2 µm. The SU8-coated wafer was baked at 65 °C for 1 min and thereafter at 95 °C for 3 min. The SU8-layer was exposed to UV light thrice (each exposure, 22.4 mW/cm², 1.7 sec) using a mask aligner (MA-20, Mikasa). The photomask for the observation channels was used in this step. The wafer was post-baked at 65 °C for 1 min and 95 °C for 3 min after the exposure and developed with the SU8 developer. The wafer was rinsed with isopropanol (Wako) and air-dried.

Next, we spin-coated SU8-3025 (MicroChem) with the target height of 20 µm. The pre-bake was performed at 65 °C for 3 min and thereafter at 95 °C for 7 min. The SU-8 layer was again exposed to UV light (22.4 mW/cm², 30 s) with the photomask for the main trench. The wafer was post-baked at 65 °C for 3 min and 95 °C for 10 min and developed with the SU8 developer. The wafer was rinsed with isopropanol and air-dried.

The Part A and Part B of polydimethylsiloxane (PDMS) resin (Sylgard 184 Silicone Elastomer Kit, Dow Corning) were mixed at the ratio of 10:1 and poured onto the SU-8 mold placed in a tray made of aluminum foil. After removing the air bubbles under decreased pressure, the PDMS was cured at 65 °C for 1 hr. We cut out a PDMS block containing the channel structures and punched holes at both ends of the main trench. The PDMS block was washed with isopropanol and heated at 65 °C for 30 min.

We washed the coverslips (thickness: 0.13–0.17 mm, 24 × 60 mm, Matsunami) by sonication with 10-fold-diluted Contaminon solution (ContaminonR○ LS-II, Wako) for 30 min, with 99.5% ethanol (Wako) for 15 min, and with 0.8 M NaOH solution (10-fold diluted 8 M NaOH (Wako)) for 30 min. The coverslips were rinsed with milliQ water by sonication after each washing step. The coverslips were dried at 140 °C for 1 hr.

We exposed the PDMS block and the coverslips to oxygen plasma using a compact etcher (FA-1, SAMCO) and bonded them together at 65 °C for 5 min. After bonding, we inserted the silicone tubes (inner diameter: 1 mm, outer diameter: 2 mm, Tigers Polymer Corporation) into the holes and smeared a small amount of pre-cured PDMS. We cured the PDMS at 65 °C overnight to fix the tubes into the holes tightly.

## Sample preparation and experimental procedures for single-cell time-lapse experiments

Cells stored as a glycerol stock were inoculated in the LB broth containing 50 µg/mL of Amp and cultured at 37 °C overnight with shaking (200 rpm). 200 µL of overnight culture was centrifuged at

21,500×g for 3 min. The supernatant was discarded, and the cells were resuspended in 1.5 mL of the M9 medium. This culture was diluted to $OD_{600}$ = 0.01 in the M9 medium containing 50 µg/mL of Amp, and incubated with shaking for 5–6 hr at 37 °C. The cell culture was again spun at 2350×g for 5 min using a centrifuge (CT6E, himac, Hitachi), and the cells were resuspended in 200 µL of the M9 medium.

Before loading the cells, the growth channels and the main trench in the mother machine device were washed by flowing 0.5 mL of 99.5% ethanol, M9 medium, and 1% (w/v) bovine serum albumin (BSA, Wako) solution sequentially using a syringe. After washing, we introduced the cell suspension into the device and let the cells enter the growth channels by placing the device at 37 °C for 1–2 hr without flow in a dark room. After confirming that the cells were trapped in many growth channels, we started flowing the M9 medium containing 0.1 mM IPTG and 0.1% (w/v) BSA at 2 mL/hr. For experiments with continuous Cp exposure, we also added 15 µg/mL or 30 µg/mL of Cp to the flowing medium from the beginning. We initiated time-lapse image acquisitions after overnight cultivation in the mother machine. We illuminated blue light for 30 min at pre-determined timings in all single-cell time-lapse experiments. The medium flow rate was maintained at 2 mL/hr and increased to 5 mL/hr for 30 min once or twice a day to prevent the formation of cell aggregates in the main trench. In Cp removal experiment, the flowing medium was switched to M9 medium containing 0.1 mM IPTG and 0.1% (w/v) BSA at 2 mL/h 72 hr after blue light exposure stopped.

## Time-lapse image acquisitions and analysis

Time-lapse image acquisitions were performed with ECLIPSE Ti fluorescent microscope (Nikon) equipped with 100× oil immersion objective lens (Plan Apo $\lambda$, NA 1.45, Nikon), digital CMOS camera (ORCA-frash, Hamamatsu Photonics), and LED light source (DC2100, Thorlabs) for fluorescence excitation. For acquiring phase-contrast images, the transmitted light was illuminated for 20 ms during *mcherry-cat* gene deletion experiments and for 50 ms during experiments performed with ribosome reporter strains through neutral density and red filters to avoid unintended activation of PA-Cre. Fluorescence images were acquired with appropriate filter cubes (YFP HQ (Nikon) for mVenus and Texas Red (Nikon) for mCherry). In the *mcherry-cat* gene deletion experiments, mCherry fluorescence was obtained every 2 min with excitation of 500 ms. The mCherry and mVenus fluorescence images were acquired every 10 min with the excitation of 100 ms in the experiments with ribosome reporter strains. IPTG was removed from the flowing media after blue-light illumination to avoid unintended gene deletion.

Blue-light illumination on the microscope for gene deletion experiments was regulated by an independent controller. We used a tape LED (7.2 mW, wave length:464~474 nm, 60 LED/1 m, LED PARA-DISE) as the blue-light source and surrounded the mother machine device on the microscope stage with this tape LED. The LED light was switched on and off with a timer (REVEX).

Image analysis was performed using ImageJ Fiji (http://fiji.sc/). Image registration was performed by HyperStackReg described in MoMA Macro (*Kaiser et al., 2018*). Cell segmentation was semi-automated and retouched manually using iPad Pro Sidecar. Semi-automated segmentation and tracking Macro reported previously (*Hashimoto et al., 2016*) were used in this study. The data from image analysis were further analyzed using Python 3 (https://www.python.org/) with some general packages, including NumPy, SciPy, pandas, Matplotlib, seaborn, FastDTW, and JupyterLab.

The transitions of mCherry-CAT fluorescence intensities and elongation rates in single-cell lineages were classified using hierarchical clustering in *Figure 5—figure supplement 2*. The full-length single-cell lineage data were used in this analysis. The similarity of each pair of time series was measured by dynamic time warping (*Salvador and Chan, 2007*). Hierarchical clustering is agglomerative; the averaged dynamic time warping was used when similarity to another time series or cluster was evaluated.

## Cell sampling from the mother machine and whole-genome sequencing

After time-lapse observations performed with the mother machine, we switched the flowing media to the M9 medium containing 0.1% (w/v) BSA and no Cp. We recovered the growth of resistance-gene-deleted cell lineages by culturing them without Cp for more than 6 hr. Subsequently, we collected the media flowing out from the mother machine in a 1.5 mL tube. The $OD_{600}$ of the collected cell suspension was measured with a spectrometer (UV-1800, Shimadzu). The cell suspension was diluted serially with the M9 medium to obtain a cell density corresponding to $OD_{600} = 3.3 \times 10^{-9}$. A total of 200 µL of

diluted cell suspension was taken in each well of 96-well plates. The number of cells in the 200 μL of cell suspension $k$ is expected to follow a Poisson distribution $P(k) = \lambda^k e^{-\lambda}/k!$ with the mean $\lambda = 0.54$. Therefore, $P(0) = 0.58$, $P(1) = 0.31$, $P(2) = 0.085$, and $P(k \geq 3) = 0.018$. The 96-well plates were incubated with shaking at 37 °C for two nights. We performed PCR on samples with high turbidity in wells to confirm the presence or absence of the *cat* gene. We selected both *cat*-positive and *cat*-negative cellular populations from the 96-well plates and cultured them in test tubes overnight at 37 °C. We stored these samples at –80 °C for further analyses.

For the measurement of mCherry-CAT fluorescence intensities of the selected samples (*Figure 3— figure supplement 3*), cells stored as a glycerol stock were inoculated in the M9 medium containing 50 μg/mL of Amp and cultured at 37 °C with shaking (200 rpm) overnight. 10 μL of overnight culture was inoculated in the M9 medium containing 50 μg/mL of Amp and incubated with shaking for 4 hr at 37 °C. A total of 0.3 μL of cell culture was placed on an agar pad prepared using the M9 medium and 1.5% (w/v) agar (Wako) and covered with a coverslip. Image acquisitions of the samples were performed with ECLIPSE Ti fluorescent microscope (Nikon) equipped with 100× oil immersion objective lens (Plan Apo $\lambda$, NA 1.45, Nikon), digital CCD camera (ORCA-R2, Hamamatsu Photonics), and LED light source (DC2100, Thorlabs) for fluorescence excitation. For acquiring phase-contrast images, the transmitted light was illuminated for 50 ms through neutral density. The mCherry fluorescence images were acquired with Texas Red filter cubes and an exposure time of 500 ms.

For the measurement of whole-genome sequencing of the selected samples, cells stored as a glycerol stock were inoculated in 5 mL of the M9 medium and cultured at 37 °C with shaking (200 rpm) overnight. The $OD_{600}$ of overnight culture was measured with a spectrometer. Rifanpicin (Wako) was added to the overnight culture to the final concentration of 300 μg/mL. After 3 hr of incubation, the genomes of these samples were extracted with DNeasy Blood and Tissue kit (QIAGEN). TruSeq DNA PCR Free kit (Illumina) was used for the whole-genome sequencing library preparation. These libraries were sequenced using NovaSeq (Illumina). Library preparation and sequencing were outsourced to Macrogen Japan (Tokyo, Japan). The sequence data were analyzed with breseq (*Deatherage and Barrick, 2014*).

We evaluated the probability that all of the five cell populations for the whole-genome sequencing were derived from growth-halted cell lineages were 5.5%. This value was obtained as $\left( \frac{(1-0.373) \times 0.693}{0.373 \times 0.913 + (1-0.373) \times 0.693} \right)^5 = 0.055$, considering the fraction of growth-restored cell lineages among the resistance-gene-deleted cell lineages (37.3%) and the proportions of the cell lienages that recovered fast growth after Cp removal (91.3% for growth-restored cell lineages; 69.3% for growth-halted cell lineages).

## Data availability

All the data obtained in this study except for high-throughput sequencing data have been deposited to a Github repository (https://github.com/YKogane/History-Dependent-Physiological-Adaptation-2021, copy archived at swh:1:rev:ddbd24c49b29ade0f667a739c3f04b5d66e12488; *Koganezawa, 2022*). High-throughput sequencing data have been deposited in the NCBI Sequence Read Archive (SRA) under BioProject accession in no. PRJNA774496.

## Code availability

All the codes used for the data analysis are available from a Github repository (https://github.com/YKogane/History-Dependent-Physiological-Adaptation-2021).

## Acknowledgements

We thank Kaito Kikuchi for the help with plasmid construction in the initial stage of this project; Stanislas Leibler, Chikara Furusawa, Kunihiko Kaneko, Tetsuya J Kobayashi, and the members of the Wakamoto Lab for discussion.

## Additional information

### Funding

| Funder | Grant reference number | Author |
|---|---|---|
| Japan Science and Technology Agency | JPMJCR1927 | Yuichi Wakamoto |
| Japan Science and Technology Agency | JPMJCR1653 | Moritoshi Sato |
| Japan Science and Technology Agency | JPMJER1902 | Yuichi Wakamoto |
| Japan Society for the Promotion of Science | 17H06389 | Yuichi Wakamoto |
| Japan Society for the Promotion of Science | 19H03216 | Yuichi Wakamoto |
| Kanagawa Institute of Industrial Science and Technology | Project Grant | Moritoshi Sato |
| Japan Society for the Promotion of Science | JP19J22506 | Yuta Koganezawa |

The funders had no role in study design, data collection and interpretation, or the decision to submit the work for publication.

### Author contributions

Yuta Koganezawa, Conceptualization, Formal analysis, Funding acquisition, Investigation, Validation, Visualization, Writing - original draft, Writing - review and editing; Miki Umetani, Investigation, Resources, Writing - review and editing; Moritoshi Sato, Funding acquisition, Resources, Writing - review and editing; Yuichi Wakamoto, Conceptualization, Funding acquisition, Supervision, Validation, Writing - original draft, Writing - review and editing

### Author ORCIDs

Yuta Koganezawa http://orcid.org/0000-0001-7720-0113
Miki Umetani http://orcid.org/0000-0002-3171-4327
Moritoshi Sato http://orcid.org/0000-0002-3095-5831
Yuichi Wakamoto http://orcid.org/0000-0002-6233-0844

### Decision letter and Author response

Decision letter https://doi.org/10.7554/eLife.74486.sa1
Author response https://doi.org/10.7554/eLife.74486.sa2

## Additional files

### Supplementary files
• Transparent reporting form

### Data availability

All the data obtained in this study except for high-throughput sequencing data have been deposited to a Github repository (https://github.com/YKogane/History-Dependent-Physiological-Adaptation-2021, copy archived at swh:1:rev:ddbd24c49b29ade0f667a739c3f04b5d66e12488). High-throughput sequencing data have been deposited in the NCBI Sequence Read Archive (SRA) under BioProject accession in no. PRJNA774496.

The following datasets were generated:

| Author(s) | Year | Dataset title | Dataset URL | Database and Identifier |
|---|---|---|---|---|
| Koganezawa Y, Umetani M, Sato M, Wakamoto Y | 2021 | History-Dependent-Physiological-Adaptation-2021 | https://github.com/YKogane/History-Dependent-Physiological-Adaptation-2021 | GitHub, History-Dependent-Physiological-Adaptation-2021 |
| Koganezawa Y, Umetani M, Sato M, Wakamoto Y | 2021 | Whole-genome sequence of *Escherichia coli* acquired from Mother Machine | https://www.ncbi.nlm.nih.gov/bioproject/PRJNA774496 | NCBI BioProject, PRJNA774496 |

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
