## [Editor Report]

This paper presents the temporal relationships between deletion of a resistance gene, introduction of antibiotic, and cell growth that are intriguing and novel. It will be of interest to researchers studying heterogeneity in antibiotic tolerance and the origins of drug resistance.

---

## [Decision Letter]

**Decision letter after peer review:**

Thank you for submitting your article "History-Dependent Physiological Adaptation to Lethal Genetic Modification under Antibiotic Exposure" for consideration by *eLife*. Your article has been reviewed by 3 peer reviewers, and the evaluation has been overseen by a Reviewing Editor and Naama Barkai as the Senior Editor. The reviewers have opted to remain anonymous.

Essential revisions:

1) It is already known that Chloramphenicol treatment leads to adaptive resistance (the MIC is higher when the cells are exposed to a gradually increasing concentration of the drug), so please articulate better the relevance of this work: what was a gap in previous studies? How does this study address it?

2) The analysis of cells recovered from the microfluidic devices allows to conclude that all mCherry negative cells have lost the cat resistance cassette (Figure S6), however only a small number (n=5) of cells were analysed to draw this conclusion. Please provide additional evidence to support the conclusion that all mCherry negative cells are also cat negative.

For example:

a. pcr-analysis of additional isolates to confirm that all mCherry negative cells are also cat negative (e.g., n=10 would reduce the chance of only having growth-halted populations to less than 1%), Or

b. Use imaging data to quantify the proportion of growth-halted cells that resume growth during the recovery phase. If this is sufficiently rare, the present genomic analysis would be sufficient.

3) Strengthen the characterization of growth restored cells: please homogenize the way growth is displayed (generation time in min, in hours, elongation rates) so as to clarify how much growth-restored cells grow. Do they return to normal growth after chloramphenicol is removed?

4) Test a broader range of chloramphenicol concentrations for Figure 2 to assess if the key effect is observed at higher doses.

5) The part of the manuscript on ribosomal stoichiometry is the most speculative part of the paper. Reviewers would like the authors to tone down the conclusions.

---

## [Author Response]

Essential revisions:1) It is already known that Chloramphenicol treatment leads to adaptive resistance (the MIC is higher when the cells are exposed to a gradually increasing concentration of the drug), so please articulate better the relevance of this work: what was a gap in previous studies? How does this study address it?

As pointed out, adaptive resistance is a phenomenon in which bacterial populations progressively enhance their resistance levels (MICs) when subject to gradual increases of antibiotics (George and Levy, *J. Bacteriol.*, 1983; Adam *et al.*, *BMC Evol. Biol.*, 2008; Sanchez-Romero and Casadesus, *PNAS*, 2014; Motta *et al.*, *PLoS ONE*, 2015). Adaptive resistance is achieved physiologically rather than genetically since the enhanced resistance is lost when the antibiotic is removed. Therefore, adaptive resistance might be related to the phenomenon identified in this study; it is plausible that growth-restoration against resistance-gene deletion and increases of MICs in adaptive resistance might be realized by some common mechanisms.

However, it is nontrivial whether bacterial cells respond to the intensification of antibiotic stress caused by internal resistance-gene deletion similarly to the stress caused by gradual increases of external antibiotic concentrations. Therefore, rather than the recapitulation of the known phenomenon, our results should be considered the first support for the hypothesis that physiological adaptation to genetic perturbations that sensitize cells to some antibiotics is feasible when bacterial cells exhibit adaptive resistance to those drugs. However, we still believe that such a conclusion should be drawn after more careful examinations of shared changes in cellular states provoked by internally and externally intensified antibiotic stress under diverse experimental conditions, which remains to be addressed in future studies.

Additionally, we remark that adaptive resistance has been mostly investigated at the population level, not at the single-cell level, in the previous studies. Consequently, it remains elusive whether only a small fraction of cell lineages in the populations acquire resistance and dominate the populations by selection or most cells can progressively enhance resistance in adaptive resistance. In this study, using the mother machine microfluidic device, we were able to quantify the fractions of cell lineages that restore growth against resistance-gene deletion and the changes of the fractions depending on the historical conditions. Such single-cell-level quantitative information would be valuable for unraveling the windows of phenotypic space and the ranges of conditions permitting physiological adaptation to antibiotics.

We now state the new insights that the results provide and their potential relevance to adaptive resistance in Discussion (L259-L281).

2) The analysis of cells recovered from the microfluidic devices allows to conclude that all mCherry negative cells have lost the cat resistance cassette (Figure S6), however only a small number (n=5) of cells were analysed to draw this conclusion. Please provide additional evidence to support the conclusion that all mCherry negative cells are also cat negative.For example:a. pcr-analysis of additional isolates to confirm that all mCherry negative cells are also cat negative (e.g., n=10 would reduce the chance of only having growth-halted populations to less than 1%), Orb. Use imaging data to quantify the proportion of growth-halted cells that resume growth during the recovery phase. If this is sufficiently rare, the present genomic analysis would be sufficient.

We thank the reviewers for drawing our attention to an important issue. As suggested, we confirmed that all mCherry negative cells were also *cat* negative by additional experiments and analyses.

a. In addition to the previous correspondence check shown in Figure 1—figure supplement 2C (previously, Figure S2C), we conducted a batch culture experiment in which the YK0083 cells were illuminated by blue light for 30 minutes under Cp exposure and plated on agar plates without Cp. We checked the deletion of *cat* gene by PCR for 46 colonies, finding that 24 colonies were *cat* negative and that all of these colonies were mCherry negative (Figure 3—figure supplement 2A). On the other hand, the other 22 *cat*-positive colonies were all mCherry positive (Figure 3—figure supplement 2A). This result confirms that mCherry fluorescence correctly reports the presence/deletion of *cat* resistance gene. Considering the frequency of growth-restored cells among resistance-gene-deleted cells (~40%), this result constitutes the firm evidence that the growth-restored non-fluorescent cells did not possess the *cat* resistance gene. We now mention this result in Results (L84-L89).

b. We also visualized regrowing dynamics of both growth-restored and growth-halted cell lineages after removing Cp in the microfluidics device (Figure 3—figure supplement 4 and Video 3). We found that 91.3% (73/80) of growth-restored cell lineages recovered fast growth after the exposure to 15 µg/ml of Cp for 72 hours. On the other hand, the recovery frequency was lower for the growth-halted cell lineages; only 69.3% (140/202) of growth-halted cell lineages could resume growth. The growth after first cell divisions were as fast as that of the non-deleted cells and was indistinguishable between the growth-restored and growth-halted cell lineages (1.07 h mean generation time; see Figure 3—figure supplement 4D). Taking the fraction of growth-restored cells among the resistance-gene-deleted cells and the proportions of growth-recovered cells after Cp removal into account, we found that the probability that all of the five *cat*-negative cell populations analyzed by whole-genome sequencing had originated from the growth-halted cell lineages was as low as 5.5% (L99-L111).

We believe that the new evidence provided by these additional experiments and analyses would more strongly support the conclusion that growth restoration of the resistance-gene-deleted cells was achieved physiologically. We now explain these new results in the Results section along with new figures and video (Figure 3—figure supplement 2A, Figure 3—figure supplement 4, and Video 3).

3) Strengthen the characterization of growth restored cells: please homogenize the way growth is displayed (generation time in min, in hours, elongation rates) so as to clarify how much growth-restored cells grow. Do they return to normal growth after chloramphenicol is removed?

We apologize for the mixed usage of the time units in the original manuscript; we now fix the time unit to hours.

As pointed out in the comment (2) of Reviewer #1, generation time and elongation rate represent different characteristics of cellular growth. We therefore believe that showing both data is more appropriate than homogenizing the way growth is displayed. However, we noted that the previous manuscript did not include the transitions of the elongation rate of YK0083 cells after resistance-gene-deletion with the horizontal axis representing absolute time (not scaled to generations). We now show this plot in Figure 3—figure supplement 1B.

Growth restoration is more evident when the growth parameters are grouped by generations after blue light illumination. However, such representation is not applicable to growth-halted cell lineages as they stopped cell divisions only in several generations after resistance gene deletion. Therefore, the comparison of growth between growth-restored and growth-halted cell lineages is possible only in the absolute time unit, and we decided to show the transitions in both generation and absolute time.

As we wrote in our reply to Essential Revision 2, 91.3% of the growth-restored deleted cells returned to normal growth after Cp removal (Figure 3—figure supplement 4 and Video 3). Growth-halted cells also returned to normal growth, though the regrowing cell frequency was lower (69.3%) than growth-restored cells. We now explain these growing dynamics of resistance-gene-deleted cells in Results (L99-L107).

4) Test a broader range of chloramphenicol concentrations for Figure 2 to assess if the key effect is observed at higher doses.

To address this, we deleted the *mcherry-cat* resistance gene of YK0083 under the exposure of a two-fold higher concentration of Cp (i.e., 30 µg/ml). 33.1% (361/1092) of the cells illuminated by blue light lost the resistance gene, and we found that none of them restored growth, unlike the previous observation under the exposure of 15 µg/ml. This result suggests that sufficiently high concentrations of Cp can prevent adaptation to resistance-gene deletion.

Related to the comment in Essential Revision (1), this result is again reminiscent of adaptive resistance, in which gradual increases of antibiotics concentration are required to gain resistance to higher concentrations. The similarity of these two adaptive phenomena might suggest that intensification of internal antibiotic stress must be sufficiently slow for the cells to enhance their resistance physiologically.

We now mention this new experimental result at a higher concentration of Cp in the Results section (L123-L127) and discuss the possible importance of slow intensification of antibiotic stress for physiological adaptation (L265-L268).

5) The part of the manuscript on ribosomal stoichiometry is the most speculative part of the paper. Reviewers would like the authors to tone down the conclusions.

We agree that recovery of ribosomal stoichiometry still remains as correlative observations with growth restoration after resistance-gene deletion. Corrections were made to the statements that could be misinterpreted as asserting a causal relationship. (L203-L204, L223-L225, and L241-L242).